# Mechanisms that allow cortical preparatory activity without inappropriate movement

Timothy R Darlington*, Stephen G Lisberger

Department of Neurobiology, Duke University School of Medicine, Durham, United States

**Abstract** We reveal a novel mechanism that explains how preparatory activity can evolve in motor-related cortical areas without prematurely inducing movement. The smooth eye movement region of the frontal eye fields (FEF$_{SEM}$) is a critical node in the neural circuit controlling smooth pursuit eye movement. Preparatory activity evolves in the monkey FEF$_{SEM}$ during fixation in parallel with an objective measure of visual-motor gain. We propose that the use of FEF$_{SEM}$ output as a gain signal rather than a movement command allows for preparation to progress in pursuit without causing movement. We also show that preparatory modulation of firing rate in FEF$_{SEM}$ predicts movement, providing evidence against the 'movement-null' space hypothesis as an explanation of how preparatory activity can progress without movement. Finally, there is a partial reorganization of FEF$_{SEM}$ population activity between preparation and movement that would allow for a directionally non-specific component of preparatory visual-motor gain enhancement in pursuit.

## Introduction

Preparation is an important component of voluntary movement. Throughout the motor system, many neurons show modulation of firing rate that occurs during movement preparation and is linked to parameters of the ensuing movement (*Dorris et al., 1997*; *Glimcher and Sparks, 1992*; *Hanes and Schall, 1996*; *Munoz and Wurtz, 1995*; *Tanaka and Fukushima, 1998*; *Tanji and Evarts, 1976*; *Weinrich and Wise, 1982*; *Wurtz and Goldberg, 1972*; *Churchland et al., 2006a*; *Churchland et al., 2006b*; *Darlington et al., 2018*; *Hanes and Schall, 1996*; *Mahaffy and Krauzlis, 2011*; *Messier and Kalaska, 2000*). Importantly, many neurons that seem to be tied to movement still have preparatory activity, even though there is neither movement nor muscle activity during preparation. How can motor and premotor cells in the cerebral cortex and even the brainstem and spinal cord (*Glimcher and Sparks, 1992*; *Munoz and Wurtz, 1995*; *Prut and Fetz, 1999*) show preparatory activity without causing inappropriate movement?

We have addressed the general question of how preparatory activity fails to evoke movement in the smooth eye movement region of the frontal eye fields (FEF$_{SEM}$), a critical node in the neural circuit for smooth pursuit eye movements. Neurons in FEF$_{SEM}$ are seemingly tied to pursuit. Inactivation or lesioning of FEF$_{SEM}$ produces dramatic defects in smooth pursuit (*Keating, 1991*; *Shi et al., 1998*). Micro-stimulation induces smooth eye movement (*Gottlieb et al., 1993*). Strong modulation of activity occurs just before the initiation of pursuit (*MacAvoy et al., 1991*). The modulation is tuned for the direction of visual motion and eye movement and is monotonically related to their speeds (*Gottlieb et al., 1994*; *Tanaka and Lisberger, 2002a*; *Tanaka and Lisberger, 2002b*). Therefore, the activity of FEF$_{SEM}$ cells seems to play an important role in the generation of smooth pursuit eye movement. Importantly, the same cells that discharge in relation to the initiation of pursuit also show an impressive ramp increase in firing rate during fixation, at a time when there is neither visual motion nor smooth eye movement (*Mahaffy and Krauzlis, 2011*; *Tanaka and*

*For correspondence:
timothy.r.darlington@gmail.com

Competing interests: The authors declare that no competing interests exist.

*Fukushima, 1998*). The preparatory activity grows over the hundreds of milliseconds preceding visual motion and the onset of pursuit, and its amplitude encodes expectation of upcoming target speed and combines with visual-motion input in a reliability-weighted manner to determine pursuit-related responses (*Darlington et al., 2018*).

Several mechanisms have been proposed in various motor systems to explain how preparatory activity can evolve without causing movement. In the arm movement system, the current paradigm suggests that populations of neurons act as a dynamical system in which preparatory activity sets the initial conditions for the progression of movement activity (*Churchland et al., 2010*; *Churchland et al., 2012*). Importantly, preparatory population activity in primary motor cortex (M1) and dorsal premotor cortex (PMd) progresses in a way that avoids dimensions related to movement (*Kaufman et al., 2014*). It remains confined to 'movement-null' dimensions until just before movement initiation, and only then begins to progress along 'movement-potent' dimensions. Indeed, the preparatory and movement-related population responses in M1 and PMd occupy orthogonal dimensions (*Elsayed et al., 2016*). Furthermore, the use of output-null neural dimensions plays a role in both M1's initial processing of visuomotor feedback during movement as well as short-term force-field adaptation in PMd (*Perich et al., 2018*; *Stavisky et al., 2017*).

In the saccadic eye movement system, movement is prevented during preparation via the action of an all-or-none inhibitory gate created by the action of omnipause neurons in the brainstem (*Cohen and Henn, 1972*; *Keller, 1974*; *Luschei and Fuchs, 1972*). Ramps of preparatory activity in the saccadic region of the frontal eye fields fail to trigger eye movements until the collective descending activity from the frontal eye fields and superior colliculus cause the omnipause neurons to cease firing, releasing the circuits that generate movement (*Dorris et al., 1997*; *Hanes and Schall, 1996*). Furthermore, subthreshold activity in the superior colliculus possesses some motor potential when omnipause neurons are inhibited by the trigeminal blink reflex (*Jagadisan and Gandhi, 2017*). However, there is no evidence that the pursuit system uses the all-or-none gate that controls saccades (*Missal and Keller, 2002*; *Schwartz and Lisberger, 1994*).

Here we propose a third, different, mechanism for movement prevention during motor preparation. The outputs from the $FEF_{SEM}$ seem to control the gain of visual-motor transmission for pursuit (*Nuding et al., 2009*; *Tanaka and Lisberger, 2001*). Further, visual-motor gain is dialed up in preparation for an imminent smooth eye movement (*Kodaka and Kawano, 2003*; *Tabata et al., 2006*). In the present paper, we show striking parallels between the progression of preparatory activity in $FEF_{SEM}$ and the preparatory enhancement of visual-motor gain in the pursuit system. We argue that the output of $FEF_{SEM}$ dials up visual-motor gain in preparation for a behaviorally-relevant visual motion of a specific direction and speed, and that the use of $FEF_{SEM}$ output as a sensorimotor gain signal allows preparation to proceed without causing movement. We also show that preparatory activity in $FEF_{SEM}$ seems to contradict the major predictions of the dynamical system framework that has arisen from work in arm motor cortex (*Churchland et al., 2010*; *Elsayed et al., 2016*; *Kaufman et al., 2014*).

## Results

We present three primary findings to support the conclusion that $FEF_{SEM}$ preparatory activity can evolve without causing smooth pursuit eye movement because its output does not drive smooth eye movement, but rather is used as a gain signal that controls access of visual motion to the oculomotor machinery. First, we demonstrate parallel progression of $FEF_{SEM}$ preparatory activity and an objective measure of visual-motor gain. Second, we show that preparatory activity in $FEF_{SEM}$ evolves in a manner that predicts movement when there is none, precluding for the smooth pursuit eye movement system the 'movement-null' space mechanism proposed for M1 and PMd preparatory activity in the arm movement system (*Kaufman et al., 2014*). Third, we show that there is some degree of reorganization of the $FEF_{SEM}$ population response, and we link the reorganization to a directionally nonspecific component of preparatory modulation of visual-motor gain. We suggest that gain control should be considered in all movement systems as one of several possible mechanisms to allow preparatory activity without movement.

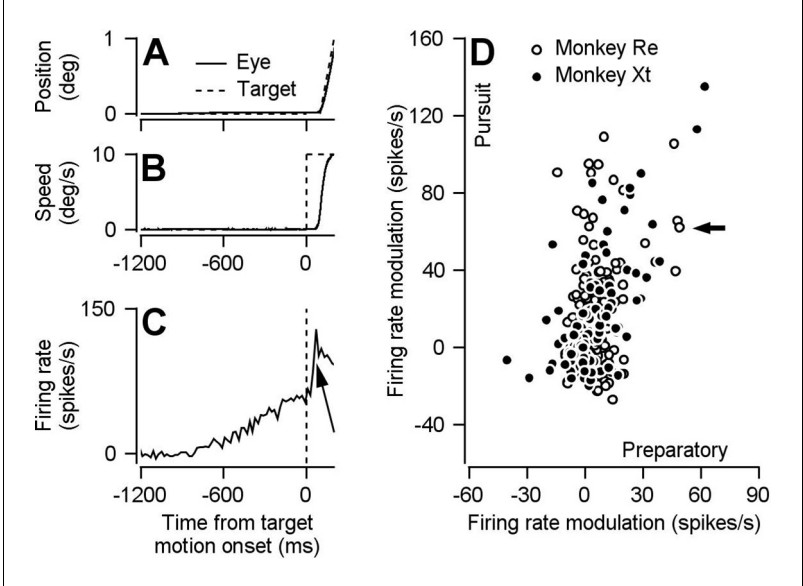

**Figure 1.** Preparatory- and pursuit-related responses in our FEF$_{SEM}$ population. (A, B) Continuous and dashed traces plot trial-averaged position (A) and speed (B) of the eye and target for one example experiment. (C) Trial-averaged firing rate as a function of time for one example neuron. The arrow points out the pursuit-related component of firing. (D) Solid and open symbols plot pursuit-related firing rate modulation as a function of preparatory-related firing rate modulation for the population of neurons recorded in monkeys Xt (Spearman's correlation coefficient = 0.55, p = 4.5 x 10$^{-10}$, n = 110 data points from 66 neurons) and Re (Spearman's correlation coefficient = 0.17, p = .02, n = 187 data points from 80 neurons). The black arrow indicates the example neuron from panel C. Pursuit firing rate modulation was calculated as the average firing rate 51-150 ms after motion onset minus the average firing rate during the last 100 ms of fixation. Preparatory firing rate modulation was calculated as the average firing rate during the last 100 ms of fixation minus the average firing rate during 151-250 ms after fixation onset. Panel D is adapted from *Darlington et al. (2018)*, except that here the analysis is performed during different behavioral conditions from the same data set and slightly different epochs are used to calculate pursuit firing rate modulation.

The online version of this article includes the following source data for figure 1:

**Source data 1.** Source data for *Figure 1*.

## FEF$_{SEM}$ preparatory activity evolves in parallel with a preparatory modulation of visual-motor gain

Neurons in FEF$_{SEM}$ are seemingly tied to the generation of smooth pursuit eye movement, yet display preparatory activity in the absence of any eye movement. We recorded single neurons in FEF$_{SEM}$ while monkeys smoothly tracked moving targets during blocks of trials that presented target motion in a single direction. The example recording in *Figure 1A–C* shows the response of a neuron with a strong ramp increase in firing rate during fixation of a stationary spot, well in advance of either (1) target motion that starts 800–1600 ms after fixation onset or (2) the initiation of eye movement. The same neuron shows a second increase in firing rate that starts after the onset of target motion, around the time of the initiation of smooth pursuit (arrow in *Figure 1C*). We observed a wide range of preparatory- and pursuit-related firing rate modulation across our FEF$_{SEM}$ population, and there was a positive relationship between preparatory- and pursuit-related firing rate modulation (*Figure 1D*, Monkey Re: Spearman's correlation coefficient = 0.17, p=0.02, n = 187 data points from 80 neurons; Monkey Xt: Spearman's correlation coefficient = 0.55, p=4.5×10$^{-10}$, n = 110 data points from 66 neurons). As explained in the Materials and methods, we ran multiple blocks of trials on most neurons using different directions of target motion, and each block provided a data point. Thus, more data points than neurons.

We hypothesized that preparatory activity in FEF$_{SEM}$ fails to cause movement because it is used downstream as a visual-motor gain signal. Gain can be dialed up by preparatory activity, but absent a movement-driving visual motion signal there will be no movement. This hypothesis is based on

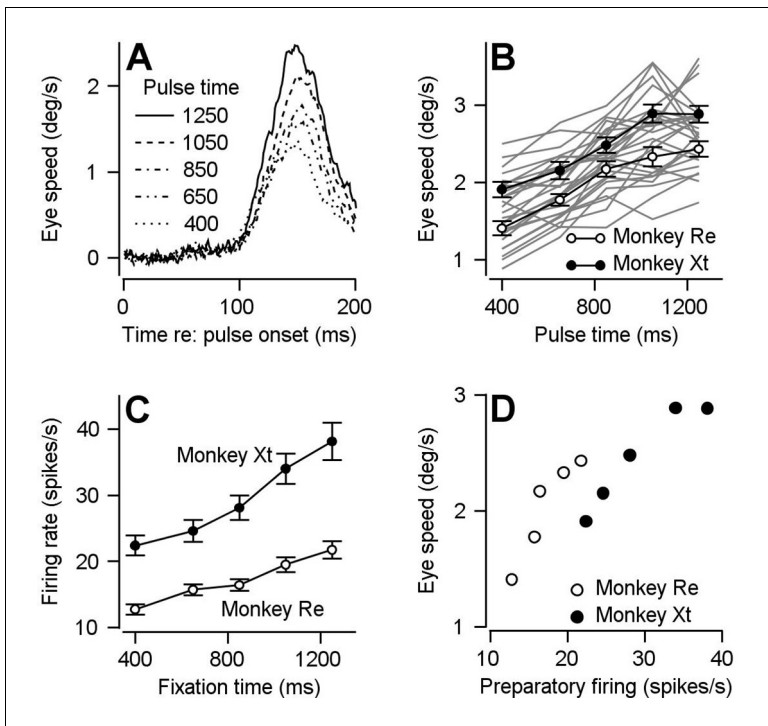

**Figure 2.** Comparison of time course of visual-motor gain and preparatory activity during fixation at various times after fixation onset. We probed visual-motor gain by measuring eye speed responses to brief target motions delivered at different times. (**A**) Different line styles plot trial-averaged eye speed traces aligned to the onset of a pulse of visual motion pulse delivered at different fixation times for one example behavioral experiment. (**B**) Filled (monkey Xt) and open (monkey Re) symbols plot eye speed averaged across experiments in response to visual motion pulses as a function of the fixation time when the pulse occurred. Error bars represent S.E.M (n = 15 behavioral experiments in Monkey Re and 12 behavioral experiments in Monkey Xt). Eye speeds were measured as the peak response in the interval from motion pulse onset to 200 ms after pulse onset. Light gray curves plot the data from individual experiments. (**C**) Filled (monkey Xt) and open (monkey Re) symbols plot average preparatory firing rate as a function of fixation time. Data were averaged across all neurons that increased their firing rate across fixation (n = 135 data points in 66 neurons for monkey Re and n = 67 data points in 45 neurons for monkey Xt). Firing rates were calculated as the average firing rate during the last 100 ms of fixation. (**D**) Filled and open symbols replot the eye speed data from panel **B** as a function of the firing rates from panel **C** for each fixation time for the two monkeys.

The online version of this article includes the following source data and figure supplement(s) for figure 2:

**Source data 1.** Source data for *Figure 2*.

**Figure supplement 1.** Preparatory firing rate for each neuron with positive preparatory modulation.

previous publications showing that FEF_SEM output during pursuit initiation is involved in controlling the gain of visual-motor transmission (*Tanaka and Lisberger, 2001*).

To test whether FEF_SEM preparatory activity could be acting as a gain signal, we conducted experiments that measured only pursuit behavior, but were carefully contrived based on knowledge of the properties of preparatory activity in FEF_SEM. We probed the state of visual-motor gain in pursuit by delivering a brief (50 ms) pulse of target motion at 5 deg/s during fixations leading up to single-direction pursuit trials. We delivered the pulse at different times during the build-up of preparatory activity and in different states of preparation. Because the test pulses of visual motion were always exactly the same and eye speed was zero at the time of the pulse, any effect of fixation time or preparatory state on the eye movement responses to the probe must reflect a change in the setting of visual-motor gain in the pursuit system.

It is important to note that the visual motion pulse experiments measured eye movement behavior only and were a direct follow-up from the analyses we performed on our neurophysiological data in FEF_SEM. We matched all experimental conditions, including the timing of fixation and visual

motion onset, in order to maximize the validity of comparisons between the behavioral and neurophysiology data sets. Further, the neurophysiological and behavioral experiments were conducted on the same pair of monkeys.

We found a series of parallels between the amplitude of preparatory activity and the setting of visual-motor gain, as probed by the size of the eye movement response to the pulse. Parallels include: (1) time course during fixation before target motion, (2) effect of variations in the expectation of target speed, and (3) direction specificity.

The behavioral assessment of visual-motor gain increases as a function of fixation time, paralleling the progression of preparatory activity in FEF$_{SEM}$. *Figure 2A* illustrates data from one example session showing larger eye speed responses at later fixation times in response to identical, but differently-timed, pulses of visual motion. The same result appeared consistently in both monkeys we tested across multiple behavioral sessions (*Figure 2B*). Modulation of FEF$_{SEM}$ preparatory activity showed a very similar time course across fixation for the overwhelming majority of neurons that had positive preparatory firing rate modulation (*Figure 2C*, *Figure 2—figure supplement 1*). We can see the tight link between visual-motor gain and preparatory firing if we plot the behavioral results

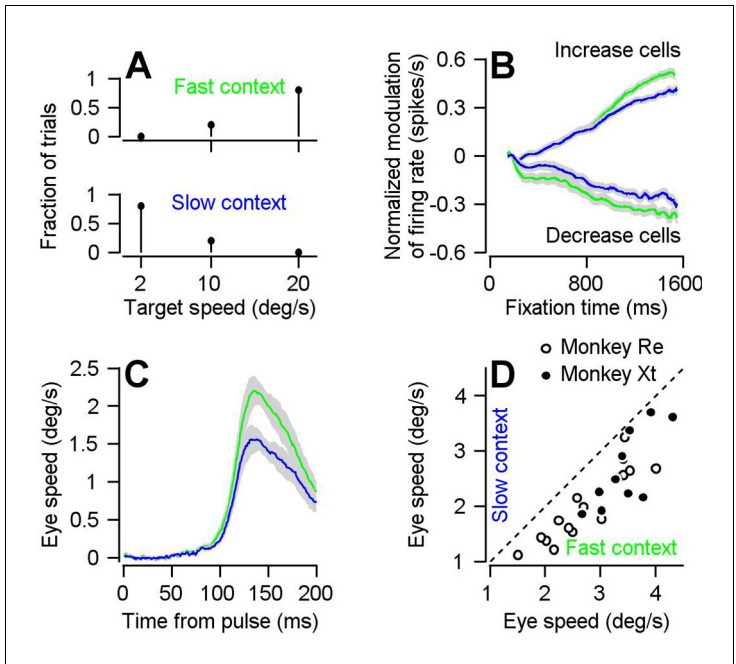

**Figure 3.** Parallel effects of expected target speed on visual-motor gain and preparatory neural activity. (**A**) Top and bottom panels plot the blend of target speeds during the fast- and slow-contexts used to control the expectation of target speed. (**B**) Green and blue traces show results for the fast- versus slow-context and plot the preparatory firing rate as a function of time during fixation, averaged separately across neurons that exhibit firing rate increases and decreases across fixation. Error-bands represent S.E.M (increase cells: n = 191 samples from 111 cells, decrease cells: n = 106 samples from 63 cells). Preparatory modulation was calculated for each sample as the average firing rate during the last 100 ms of fixation minus the average firing rate during 151–250 ms after fixation onset. Samples with positive and negative modulation were classified as increase and decrease cells. According to this classification, firing rate increased during fixation for 82 cells, decreased during fixation for 34 cells, and increased during fixation for some directions and decreased for other directions in 29 cells. (**C**) Green and blue traces show data obtained during the fast and slow contexts and plot the experiment-averaged eye speed responses aligned to the onset of the visual motion pulse. Error-bands represent S.E.M. (**D**) Open (monkey Re) and solid (monkey Xt) data symbols plot the trial-averaged peak eye speed response to visual motion pulses during the slow-context as a function of that during the fast context. Each symbol shows data for a single experiment (n = 15 behavioral experiments in Monkey Re and n = 11 behavioral experiments in Monkey Xt). Dashed line indicates unity.

The online version of this article includes the following source data for figure 3:

**Source data 1.** Source data for *Figure 3*.

from *Figure 2B* against the preparatory firing from *Figure 2C* for each fixation time and each monkey (*Figure 2D*). The relationship is fairly linear with a positive correlation between neural responses and behavior across our population of neurons (slopes were computed for each neuron using linear regression: monkey Re mean = 0.073 degrees/spike, p=2.34×10$^{-11}$, n = 135 data points from 66 neurons monkey Xt mean = 0.074 degrees/spike, p=2.76×10$^{-11}$, n = 67 data points from 45 neurons).

The behavioral assessment of visual-motor gain varies in parallel with preparatory activity when we change the expected target speed by changing the blend of target speeds in a block of trials. We showed previously that the amplitude of FEF$_{SEM}$ preparatory activity encodes an expectation of target speed based on experience in blocks of 50 trials with different blends of target speed (*Figure 3A*; *Darlington et al., 2018*). Preparatory firing rate modulation was larger in magnitude (*Figure 3B*) during the 'fast-context' (green traces, 80% of trials at 20 deg/s and 20% at 10 deg/s) compared to during the 'slow-context' (blue traces, with 80% of trials at 2 deg/s and 20% at 10 deg/s). Based on behavioral assessment using pulses of visual motion during fixation, visual-motor gain during the preparatory period also is enhanced in the fast context compared to the slow context. Eye velocity responses were larger during the fast-context versus the slow-context when we presented the same visual motion pulses during fixation at different times (green versus blue in

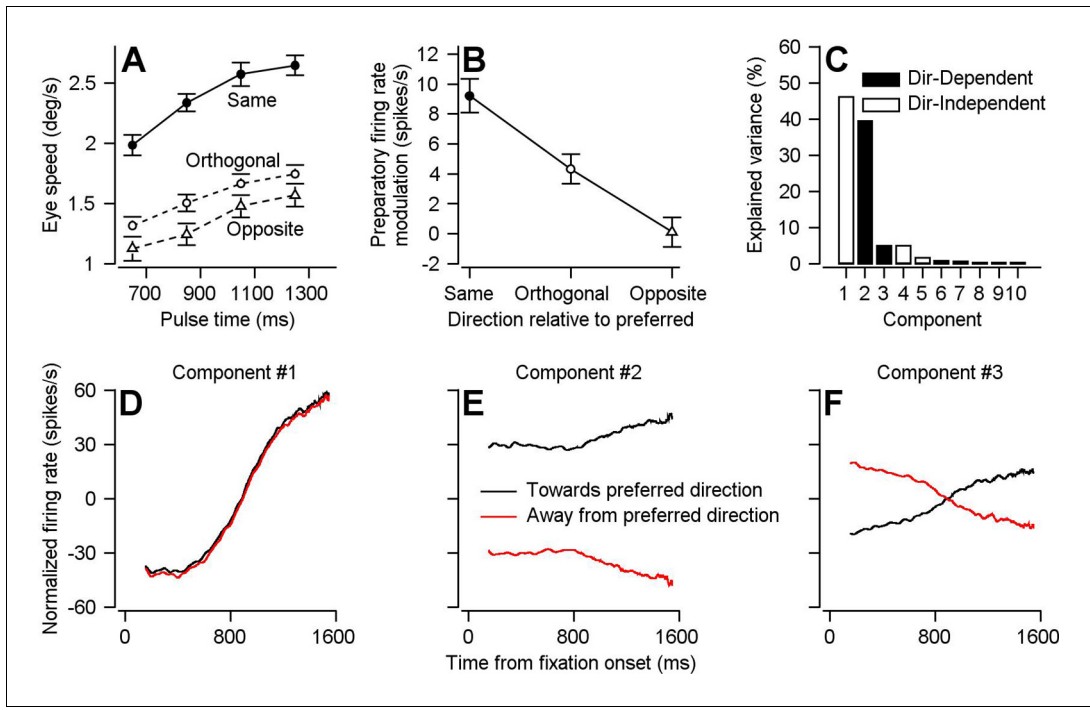

**Figure 4.** Parallels in directional specificity of eye speed responses to visual motion pulses and FEF$_{SEM}$ preparatory activity. (**A**) Filled circles, open circles, and open triangles plot eye speed in response to visual motion pulses delivered during fixation in the same, orthogonal, and opposite direction as the expected pursuit direction. Data are averaged across experiments for both monkeys and error-bars plot S.E.M (n = 30 behavioral experiments across two monkeys). Fixation times were binned across 200 ms and plotted on the x-axis at the value in the middle of the bin. (**B**) Symbols plot average preparatory modulation of firing rate versus the direction of the experiment relative to the preferred direction of the cell. Error-bars represent the S.E.M. (Same: n = 119 samples across 97 neurons, Orthogonal: n = 111 samples across 79 neurons, Opposite n = 67 samples across 62 neurons). (**C**) Bar graph plots the percentage of variance explained by each of the 10 components obtained by demixed principal component analaysis (dPCA). Filled and open bars designate direction-dependent versus direction-independent variance contributions to each component. (**D**), (**E**), (**F**) Black and red curves plot the projection of FEF$_{SEM}$ preparatory activity during single-direction blocks of trials that were toward or away from the preferred direction of each cell onto the first (**D**), second (**E**), and third (**F**) components obtained used dPCA.

The online version of this article includes the following source data for figure 4:

**Source data 1.** Source data for *Figure 4*.

*Figure 3C*, scatter plot in *Figure 3D*). In the fast-context versus slow-context, the mean eye velocity responses to the pulse were $3.02 \pm 0.71$ deg/s (SD) versus $2.27 \pm 0.73$ deg/s (SD), and the difference was statistically significant (p=$8.30\times10^{-6}$, paired, two-sided Wilcoxon signed rank test, n = 26 behavioral experiments in two monkeys, z = 4.46, Cohen's d = 1.06).

The directional specificities of preparatory modulation of visual-motor gain and FEF$_{SEM}$ preparatory activity vary in parallel. To assess the directional specificity of preparatory visual-motor gain modulation, we delivered visual motion pulses during fixation again, but now in four different directions relative to the consistent direction of the pursuit target motion in a given single-direction block: 0 deg, 'same-direction';$\pm$90 deg, 'orthogonal-direction'; 180 deg, 'opposite-direction'. Across the full duration of fixation, eye speed responses were largest for same-direction pulses, intermediate for orthogonal-direction pulses, and smallest for opposite-direction pulses (*Figure 4A*). For all directions, eye speed responses increased as a function of time during fixation for same-, orthogonal-, and opposite-direction pulses. We quantified the effect of fixation time by computing the slope of response amplitudes across pulse times. For the same direction, orthogonal direction, and opposite direction pulses, the mean (+/- S.D.) slopes across 30 experiments in two monkeys were $1.05 \pm 0.59$ deg/s per s, $0.73 \pm 0.40$ deg/s per s, and $0.72 \pm 0.45$ deg/s per s. The effect of fixation time on response amplitude was significant across all directions using 2-sided Wilcoxon signed rank tests: same-direction (p=$2.60\times10^{6}$, z = 4.70); orthogonal-direction (p=$1.73\times10^{-6}$, z = 4.78); and opposite-direction (p=$4.78\times10^{-6}$, z = 4.78). The differences from same-direction were statistically significant using the paired sample, 2-tailed Wilcoxon signed rank test for n = 30 experiments on two monkeys: same versus orthogonal-pulses (p=0.0093, z = 2.60, Cohen's d = 0.63); same- versus opposite-direction pulses (p=0.029, n = 30, z = 2.19, Cohen's d = 0.62). Thus, there is a greater preparatory enhancement of visual-motor gain in the expected direction of the upcoming pursuit trial. Further, the increase in response amplitude over fixation time for all directions of pulses indicates a degree of directionally non-specific preparatory enhancement of visual motor gain.

FEF$_{SEM}$ preparatory activity also is related to the expectation of target direction created by adjusting the preponderance of target directions in a block of trials. We divided our FEF$_{SEM}$ population into three groups according to the relative difference between the preferred direction of each cell and the direction of the experiment: same direction, $0 \pm 45$ deg; orthogonal, $+90 \pm 45$ deg and $-90 \pm 45$ deg; and opposite, $180 \pm 45$ deg. Consistent with a larger behavioral preparatory enhancement of visual-motor gain in the expected direction, FEF$_{SEM}$ neurons display the largest modulation of firing rate during fixations leading to target motion towards their preferred direction (*Figure 4B*). Preparatory firing rate modulation was smallest for upcoming target motions that were opposite to the preferred direction and intermediate for upcoming target motions that were orthogonal to the preferred direction.

We find even more compelling neural correlates of the behavior shown in *Figure 4A* using a semi-supervised variant of principal component analysis: demixed principal component analysis (dPCA) (*Kobak et al., 2016*). The primary difference is that dPCA retains knowledge of experimental conditions when finding components, allowing for identification of different patterns of population activity that are independent of, or dependent on, experimental condition. We performed dPCA on the preparatory responses of a subset of our FEF$_{SEM}$ population (n = 63 neurons) from which we were able to collect data during experiments toward *and* away from the preferred directions of each neuron. Thus, the sole condition in our dPCA is the direction of the ensuing target motion and eye movement. Our aim was to identify 'direction-dependent' and 'direction-independent' components to test the idea that FEF$_{SEM}$ contributes to the directionally specific and non-specific components of preparatory visual-motor gain enhancement shown in *Figure 4A*.

FEF$_{SEM}$ population activity can fully account for the directional features of preparatory visual-motor gain enhancement. The first three components obtained via dPCA account for ~90% of the variance of our FEF$_{SEM}$ population preparatory response (*Figure 4C*). Each of the three primary components predict features of the behavioral data shown in *Figure 4A*. Component one is direction-independent. The projections of FEF$_{SEM}$ preparatory population activity onto component one both increase across fixation for impending target motion toward and away from the preferred direction (*Figure 4D*). The projections onto component one are consistent with the directionally nonspecific enhancement of visual-motor gain observed for the orthogonal and opposite-direction visual motion pulses. Component two is direction-dependent. There are two primary differences between the projections of FEF$_{SEM}$ population activity onto component two during preparation for target motion

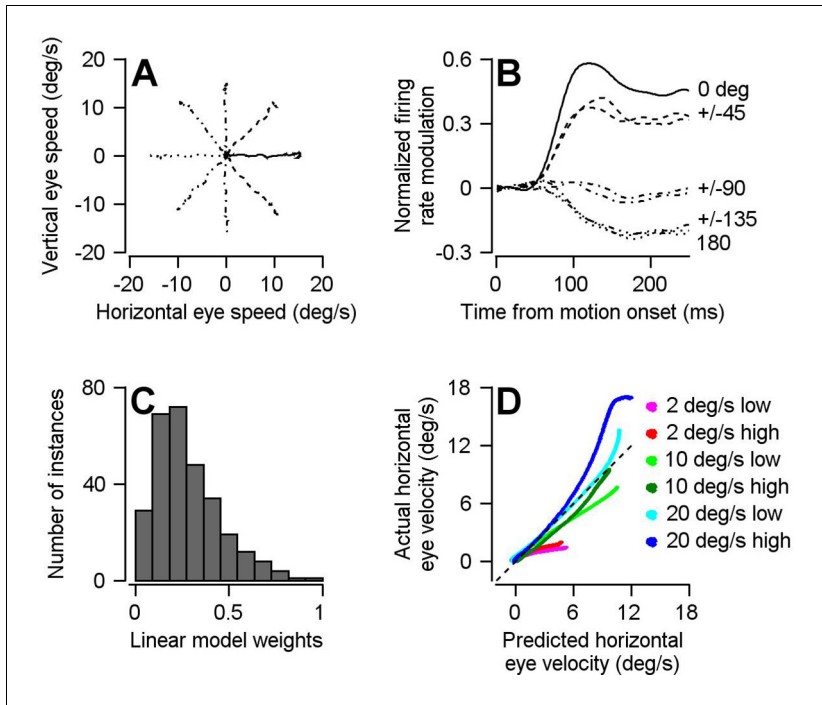

**Figure 5.** The optimal linear model for converting the population activity in FEF$_{SEM}$ into eye velocity. (**A**) Different line styles plot trial-averaged horizontal versus vertical eye velocity for smooth pursuit eye movement to 15 deg/s target motion in eight different directions (4 cardinal and four oblique directions). (**B**) Traces with the same line styles as in **A** plot the population trial-averaged firing rate after rotating directions to align the preferred direction of each cell at 0 deg (n = 146 FEF$_{SEM}$ neurons collected in two monkeys). (**C**) Histogram plots the distribution of calculated linear model weight magnitudes across our population of FEF$_{SEM}$ units. (**D**) Cross validation of the optimal linear model. The curves plot the horizontal eye velocity predicted during the single-direction block of trials by the model versus actual trial-averaged horizontal eye velocity. Different colored traces differentiate among 6 combinations of target contrast and speed.

The online version of this article includes the following source data for figure 5:

**Source data 1.** Source data for *Figure 5*.

toward and away from the preferred direction (*Figure 4E*). (i) The projections are offset from each other with the projections for toward and away motion sitting above and below each other, respectively. The offset between the projections is consistent with the behavioral finding of an offset between the eye speed responses to brief target motions in the same versus opposite direction from expected target motion: there is a substantial difference in eye speed responses to the same and opposite visual motion pulses even at the earliest fixation times we tested. (ii) The projection for motion toward the preferred direction increases and that for motion away from preferred projection decreases during the latter half of the fixation period. Component three also is direction dependent. The projection of FEF$_{SEM}$ population activity onto component three also shows a ramp increase for impending target motion toward the preferred direction and a ramp decrease for impending target motion away from the preferred direction. The increase and decrease of the projections onto components 2 and 3 are consistent with the larger enhancement of visual-motor gain across fixation time for the same versus opposite direction visual motion pulses. In conclusion, the neural basis for the behavioral preparatory enhancement of visual-motor gain can be fully reconstructed from the first 3 dPCA components of the FEF$_{SEM}$ preparatory population response.

## Modulation of FEF$_{SEM}$ preparatory activity occurs along movement-potent dimensions

Studies of arm movement have shown that population preparatory activity in M1 and PMd evolves in a manner that does not predict movement (*Kaufman et al., 2014*). Movement is prevented during

preparation because the population activity resides in 'movement-null' neural dimensions. We now test this idea on our population of FEF$_{SEM}$ neurons. A priori, such a mechanism could generalize to the smooth pursuit eye movement system, especially since humans and monkeys can initiate smooth pursuit in the absence of visual motion under certain conditions (*Freyberg and Ilg, 2008*; *de Hemptinne et al., 2006*; *Kowler et al., 2019*). Even if the output of FEF$_{SEM}$ truly is used as a gain signal, noise immunity might be enhanced if population activity resided in an output null state until there is a believable visual motion signal to drive pursuit initiation. Two mechanisms for preventing unwanted movement might be better than one.

We established the movement-potent dimensions for our sample of FEF$_{SEM}$ neurons from blocks of trials where the monkey pursued a patch of dots that moved in each of 8 randomly chosen directions (*Figure 5A*). The population average of FEF$_{SEM}$ firing across our full sample of neurons was strongly tuned to the direction of target motion and pursuit (*Figure 5B*), with a large increase in firing rate in the 'preferred direction' of the cell, a smaller increase when motion was at +/- 45 degrees from the preferred direction, no modulation for visual motion orthogonal to the preferred direction, and decreases in firing rate for +/- 135 degrees and 180 degrees from the preferred direction. We represented the movement-potent dimension for our population by computing the optimal linear model that related the pursuit-related firing of our sample of FEF$_{SEM}$ neurons to the eye movements evoked during the initiation of pursuit in eight directions. The linear model optimized two weights for each neuron: one related the neuron's firing rate to horizontal eye velocity and the second related the neuron's firing rate to vertical eye velocity. Our fitting procedure assigned a continuous distribution of weight magnitudes to the members of our population (*Figure 5C*), showing that the model used the entire FEF$_{SEM}$ population and did not simply pick out a few particularly informative cells. The model worked well on data from pursuit trials that were not used to compute the weights: target motion at three different speeds (2, 10, and 20 deg/s) and two different contrasts (12% and 100%) from a separate block of trials that presented visual motion in a single, repeated direction. It produced fairly accurate predictions for horizontal eye velocity (*Figure 5D*, slope = 1.09, r$^2$ = 0.85) and accurately predicted zero vertical eye velocities (data not shown). Because a model based on the responses to 8 directions of motion was able to predict well the responses in a different block of trials with a single direction of motion, we conclude that it is valid to use a single model across different blocks of behavior.

FEF$_{SEM}$ preparatory activity does not avoid movement-potent neural dimensions. The optimal linear model predicts movement when there is none, during fixation and preparatory activity. When we passed the trial-averaged population preparatory firing rate modulation from a single-direction block of pursuit trials through the model, we obtained a non-zero prediction for change in eye velocity across fixation (*Figure 6A and B*). Importantly, the output of the linear model predicts eye movement in a direction that is appropriate for the upcoming target motion according to how neurons across experimental sessions were combined into a pseudo-population (see below): a large positive horizontal component and relatively smaller vertical component.

We obtained similar results when we asked whether we could verify the failings of the movement-null hypothesis for FEF$_{SEM}$ in neural responses from single trials. We generated 1000 single-trial pseudo-population responses by randomly sampling preparatory firing rate modulations from each cell for fixation-duration matched trials across our population. We passed each pseudo-population response into the optimal linear model to predict the change in horizontal and vertical eye speed across fixation, and computed eye direction and speed during fixation on each 'trial'. Consistent with the findings of *Figure 6A and B*, the single trial analysis predicts non-zero eye velocity with a significant projection in the direction of the subsequent pursuit trial (*Figure 6C and D*). The mean predicted direction was −1.3 deg with a 95% bootstrap interval of −65.8 to +71.8 deg; the mean predicted eye speed was 2.9 deg/s. Thus, for the motor system we study, the data do not fulfill one important prediction of the null-space hypothesis: the optimal linear model for pursuit initiation predicts significant non-zero eye velocities driven by preparatory activity.

We made a number of modifications in adapting the analyses performed in *Kaufman et al. (2014)* to our data. These modifications were made due to a key difference in our behavioral paradigm: the upcoming direction of arm movement was explicitly cued in *Kaufman et al. (2014)*; the direction of the upcoming visual motion was not explicitly cued in our task. Therefore, only the single-direction block allows for accurate preparation, as the monkey is able to anticipate the upcoming visual motion. However, the eight-direction block provides a more rigorous mapping of FEF$_{SEM}$

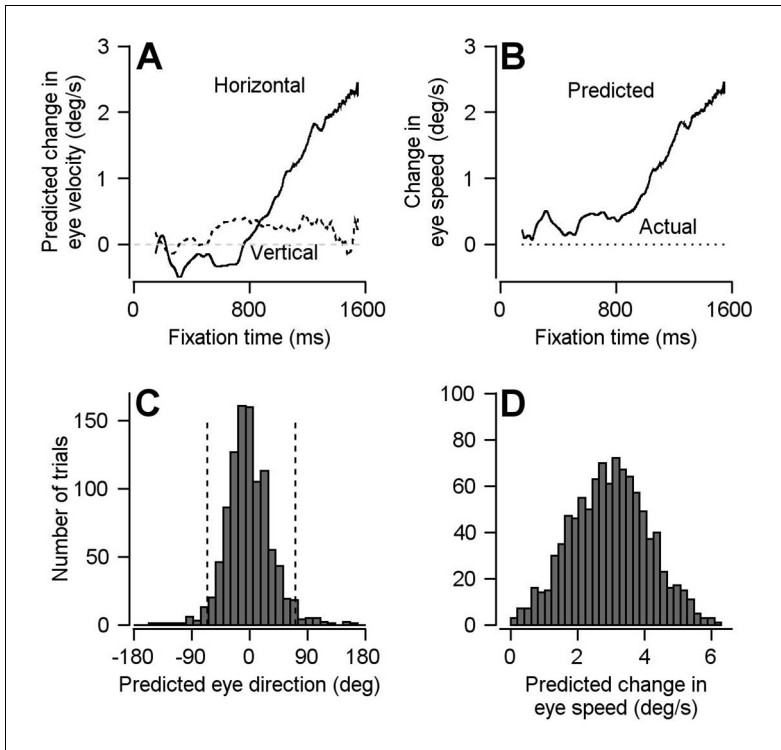

**Figure 6.** Result of using optimal linear model to transform preparatory activity in FEF_SEM into a prediction of eye velocity during preparation for movement. (**A**) Solid and dashed traces plot the linear model's predictions for horizontal and vertical eye velocity during fixation. (**B**) Solid and dotted traces plot the linear model's prediction for eye speed during fixation and the actual eye speed during fixation. (**C, D**) Histograms plot the distributions of predicted eye movement direction and speed during fixation from 1000 bootstrapped iterations of preparatory modulation of activity for single trials. The dashed lines in **c** demarcate the middle 95% of the distribution.

The online version of this article includes the following source data and figure supplement(s) for figure 6:

**Source data 1.** Source data for *Figure 6*.
**Figure supplement 1.** Findings based on the optimal linear model are not the result of oversampling data from neurons' preferred directions.
**Figure supplement 1—source data 1.** Source data for *Figure 6—figure supplement 1*.
**Figure supplement 2.** Findings based on the optimal linear model are not the result of including multiple data points from each neuron in our pseudo-population.
**Figure supplement 2—source data 1.** Source data for *Figure 6—figure supplement 2*.
**Figure supplement 3.** Repeating the optimal linear analysis in a manner more congruent to that used in *Kaufman et al. (2014)* produces the same conclusions.
**Figure supplement 3—source data 1.** Source data for *Figure 6—figure supplement 3*.

pursuit-related firing rate to eye velocity. Thus, we chose to find the optimal mapping of FEF_SEM firing rate to eye movement in the 8-direction block and test preparatory activity in the single-direction block.

Our strategy was then to create as large of a population response as we could, without excluding data points or neurons. We also did not wish to bias our data collection for the single-direction experiment towards any part of the direction tuning curve of our recorded neurons, for example by only running the single-direction experiment in the preferred direction of our neurons. To this end, we ran multiple single-direction blocks of trials for each neuron that we recorded using target directions that were different from each other and at various points on each neuron's tuning curve. We then rotated each neuron's direction tuning curves while fitting the linear model to the 8-direction block in a manner that made it seem like all of the single-direction experiments were run in the same direction (0 degrees, rightward). We included multiple single-direction blocks from the same neurons

as if they were independent under the relatively safe assumption that there is a different neuron with a tuning curve that is naturally aligned in the manner that we rotated our data.

We performed several control analyses to confirm that none of our analytical choices produced our results. First, the findings of *Figure 6* are not a product of oversampling the preferred directions of our FEF_SEM cells during the single-direction experiments (*Figure 6—figure supplement 1*). Second, we obtained the same results when we used only a single block of trials from each neuron so that our conclusions are strengthened statistically but do not depend on the use of multiple data points from each neuron as independent units (*Figure 6—figure supplement 2*). Third, a supplementary analysis that is more congruent with the analysis performed in *Kaufman et al. (2014)*, and that uses a model based on the single-direction block, arrives at the same conclusions (*Figure 6—figure supplement 3*).

## Partial reorganization of population activity from preparatory- to pursuit-related dimensions

If preparatory and pursuit-related activity resided in orthogonal dimensions, then we would say that the population activity in FEF_SEM reorganized completely in the transition from preparation to action. If they resided in parallel dimensions, then we would say that they did not reorganize at all. We now show that the truth lies between these two extremes by using principal component analysis to compare the neural dimensions related to preparation and movement. The existence of a partial reorganization explains why we find that preparatory activity predicts movement and still allows the operation of our system to align to some degree with the operation of the arm movement system, where there is a full reorganization.

After reducing the dimensionality of our preparatory- and pursuit-related population responses during the single-direction trial blocks, we analyze the first preparatory and pursuit PCs ($PC_{Prep1}$ and $PC_{Purs1}$), each of which captures more than 70% of the variance (*Figure 7A*), the same amount captured by the dimensionality reduction in *Elsayed et al. (2016)*. We validated $PC_{Purs1}$ by showing that the projection of pursuit-related activity onto the $PC_{Purs1}$ captures differences in both the speed of target motion and the contrast of the target: in parallel with the neural responses, the projections scaled with target speed and are larger in magnitude and shorter in latency for high- compared to low-contrast targets (*Figure 7B*). As expected, given the results in the previous section using the optimal linear model, the projection of preparatory activity onto $PC_{Purs1}$ shows a ramp increase across fixation (*Figure 7D*). The main difference between the preparatory and pursuit projections is that the preparatory projection ramps up more slowly to a smaller magnitude compared to the pursuit projections.

We demonstrate a partial reorganization of population activity between preparation and pursuit-related activity by computing the angle between $PC_{Prep1}$ and $PC_{Purs1}$. We used a bootstrap approach that sampled trial-averaged neurons from our population, independently found $PC_{Prep1}$ and $PC_{Purs1}$ for the sampled population, and calculated the angle between them. We used a sampling procedure that created 1000 simulated population responses, each with 160 units having a uniform representation of preferred directions (see Materials and methods and Supplementary Figure 1 for more details). We find that the mean angle between $PC_{Prep1}$ and $PC_{Purs1}$ is 65.2 degrees and is significantly different from both 0 and 90 (*Figure 7C*, p<0.001, two-sided bootstrap statistics, 95% bootstrap interval = 61.4–69.2 deg, n = 1000 iterations). Therefore, preparatory- and pursuit-related neural dimensions are neither aligned nor completely orthogonal in our population of FEF_SEM neurons. As with the linear model, our findings were not influenced by our choice to treat multiple data points from each neuron as independent units in our pseudo-population (*Figure 7—figure supplement 1*).

To understand what features of the neural responses caused the reorganization of the population response, we examined how the PCA component loadings, defined as the weights relating each neuron's response to the principal component, changed between $PC_{Prep1}$ and $PC_{Purs1}$. The distribution of the ratios between the PCA component loadings of $PC_{Prep1}$ and $PC_{Purs1}$, generated from the bootstrap in *Figure 3B*, is significantly bimodal (*Figure 8A*, Hartigan's Dip Test, Dip statistic = 0.0794, p=0.001). One peak is at a positive ratio and a smaller peak is at a negative ratio. A positive ratio indicates that a given unit behaves in a similar manner during preparation and movement, whereas a negative ratio indicates a change in behavior between preparation and pursuit. We conclude that there are two subpopulations in FEF_SEM, and that the one with negative $PC_{Prep1}/PC_{Purs1}$ ratios may

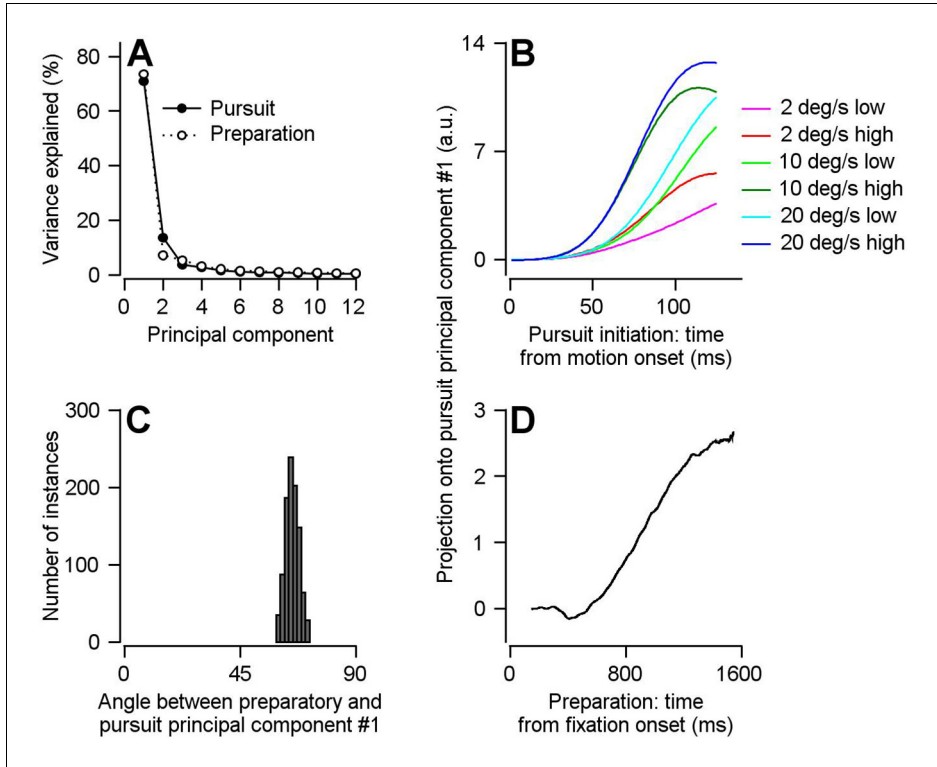

**Figure 7.** Direct assessment of preparatory- versus pursuit-related neural dimensions through comparison of principal components for preparatory- and pursuit-related population activity in FEF$_{SEM}$. (**A**) Solid and open symbols plot the percentage of the total pursuit-related and preparatory response variances explained by each principal component. Beyond principal component 3, the open symbols obscure the filled symbols. (**B**) Different colored traces plot projection of pursuit-related population firing rates in response to low-contrast and high-contrast target motion of 20, 10, and 2 deg/s onto the first pursuit principal component. (**C**) Histogram plots the distribution of the angle between the first pursuit-related and preparatory principal components across 1000 bootstrapped iterations. (**D**) Projection of population preparatory firing rates during fixation onto the first principal component of pursuit-related responses.

The online version of this article includes the following source data and figure supplement(s) for figure 7:

**Source data 1.** Source data for *Figure 7*.

**Figure supplement 1.** Findings based on the principal component analysis are not the result of including multiple data points from each neuron in our pseudo-population.

**Figure supplement 1—source data 1.** Source data for *Figure 7—figure supplement 1*.

be responsible for the partial reorganization of population activity between preparation and movement.

To test our hypothesis about the role of neurons with negative PC$_{Prep1}$/PC$_{Purs1}$ ratios, we separated our population into units with positive (Subpopulation 1) and negative (Subpopulation 2) ratios and compared a number of properties of the two subpopulations. (1) Preferred directions (*Figure 8B*): Subpopulation 1 consistently made up a larger proportion of units in our bootstrap analysis and contains a higher proportion of units with a preferred direction towards the direction of the single-direction experiment. Conversely, Subpopulation 2 contains a higher proportion of units with preferred directions away from the direction of the experiment. (2) Preparatory versus pursuit modulation: When we redid principal component analysis on the full (non-bootstrapped) data, neurons in Subpopulation 1 had statistically larger amplitudes of both preparatory- and pursuit-related modulation (compare red and black data points in *Figure 8C*; preparatory activity: p=0.019, Cohen's d = 0.34, pursuit activity: p=2.7×10$^{-12}$, Cohen's d = 0.95, 2-sided Wilcoxon rank sum tests). There is a significant positive correlation between preparatory and pursuit modulation in Subpopulation 1 (*Figure 8C*, Spearman's correlation coefficient = 0.59, p=7.8×10$^{-21}$, n = 207 samples from 130

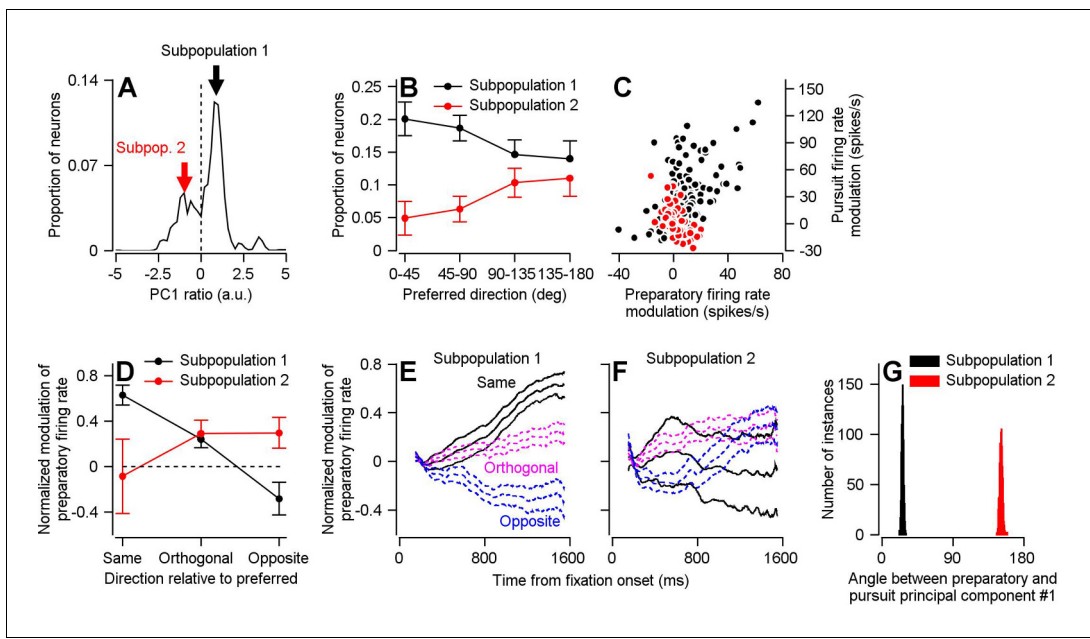

**Figure 8.** Two subpopulations of FEF_SEM neurons with different relationships between preparatory and pursuit-related activity. (**A**) The distribution of $PC_{Prep1}/PC_{Purs1}$ ratios averaged across 1000 bootstrapped iterations. The dashed line indicates a ratio of 0, the point where we split our FEF_SEM population into two subpopulations. (**B**) Black and red symbols summarize the distributions of preferred directions relative to the direction of the single-direction experiment for units in Subpopulations 1 and 2 in the bootstrap analysis. Error bars represent the middle 95% distribution obtained from our bootstrap analysis (n = 1000 iterations). (**C**) Black and red symbols plot preparatory- versus pursuit-related firing rate modulations for Subpopulations 1 and 2 in our full FEF_SEM data set (n = 207 data points from 130 neurons for Subpopulation 1 and n = 90 data points from 65 neurons for Subpopulation 2). Panel **C** plots the same data from *Figure 1D*, except here it has been separated into Subpopulations 1 and 2 (**D**) Black and red symbols plot data for subpopulations 1 and 2 and show the normalized preparatory modulation of firing rate versus the direction of the experiment relative to the preferred direction of the cell. Error-bars represent the middle 95% of the bootstrapped distribution (n = 1000 iterations). (**E**), (**F**) Black, magenta, and blue traces plot normalized modulation of preparatory firing rate as a function of time from fixation onset when experiments were run in the same, orthogonal, and opposite direction as the preferred direction of the neuron. Data were averaged separately across neurons in Subpopulations 1 (**E**) and 2 (**F**). For each direction, the middle trace is the average and the flanking traces border the middle 95% of the bootstrapped distribution, n = 1000 iterations. (**G**) Black and red histograms plot the distribution of the angle between the first pursuit-related and preparatory principal components for Subpopulations 1 and 2 across 1000 bootstrapped iterations. The online version of this article includes the following source data for figure 8:

**Source data 1.** Source data for *Figure 8*.

neurons) and a significant negative correlation in Subpopulation 2 (*Figure 8C*, Spearman's correlation coefficient = $-0.63$, p=$3.8\times10^{-11}$, n = 90 samples from 65 neurons). (3) Principal component alignment: For Subpopulation 1 alone, the first principal components of pursuit and preparatory activity are separated by an angle of only 26.2 degrees. For Subpopulation 2 alone, the angle between $PC_{Prep1}$ and $PC_{Purs1}$ is 151.2 degrees (*Figure 8G*). These facts suggest that Subpopulation 2 is responsible for the partial reorganization of population activity between preparation and movement, and that the large difference in response amplitudes between the two populations is at least partly responsible for a lack of full reorganization between preparatory- and movement-related dimensions.

The existence of two subpopulations of FEF_SEM neurons allows for a directionally-nonspecific component of preparatory visual-motor gain enhancement. In Subpopulation 1, preparatory modulation of firing rate has conventional direction tuning. It is large and positive when the upcoming target motion will be in the preferred direction of the cell, smaller when it will be orthogonal to the preferred direction, and negative when it will be opposite to the preferred direction of the cell (*Figure 8D,E*). In Subpopulation 2, preparatory modulation of firing rate has weaker and non-

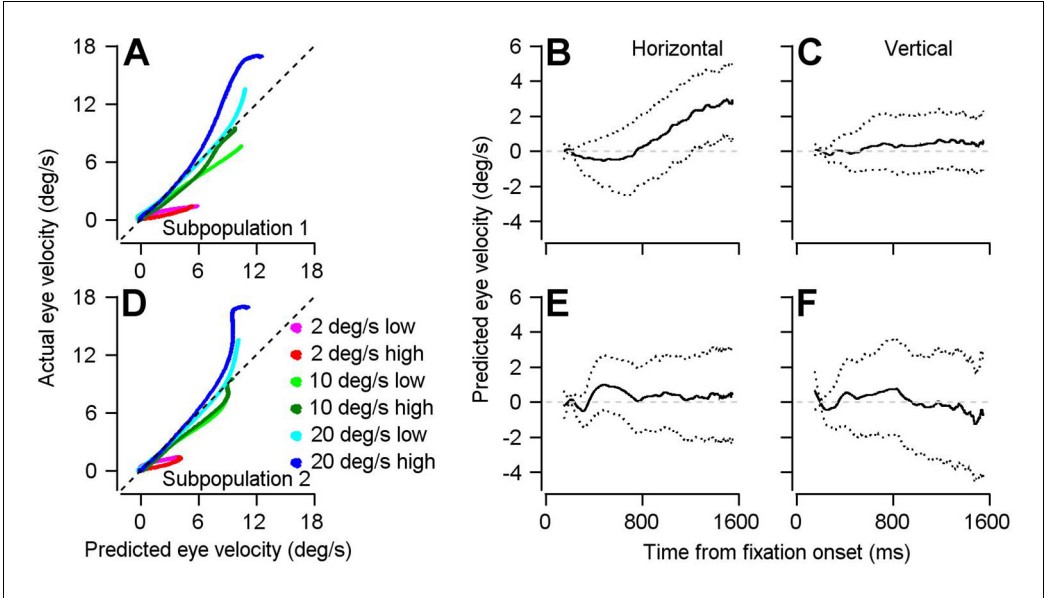

**Figure 9.** Result of using optimal linear model to transform preparatory activity in the two subpopulations into a prediction of eye velocity during preparation for movement. (**A, D**) The curves plot the horizontal eye velocity predicted during the single-direction block of trials by the model for Subpopulation 1 (**A**) and Subpopulation 2 (**D**) versus actual trial-averaged horizontal eye velocity. (**B, C, E, F**) Solid traces plot the mean of the bootstrapped linear model's predictions for horizontal (**B, E**) and vertical (**C, F**) eye velocity during fixation for Subpopulation 1 (**B, C**) and Subpopulation 2 (**E, F**).: Different colored traces in **A** and **D** differentiate among 6 combinations of target contrast and speed. Dotted lines in **B, C, E,** and **F** plot the middle 95% of the 1000 bootstrapped iterations. The online version of this article includes the following source data for figure 9:

**Source data 1.** Source data for *Figure 9*.

conventional direction tuning. Modulation is positive when the upcoming target motion is orthogonal or opposite to the preferred direction and absent when target motion will be toward the preferred direction (*Figure 8D,F*). Together, the two subpopulations can explain two features of *Figure 4A*. The preparatory activity in Subpopulation 1 explains why preparatory enhancement of visual-motor gain is largest in the direction of the upcoming target motion. The smaller preparatory activity in Subpopulation 2 might explain the directionally-nonspecific, and smaller, component of preparatory visual-motor gain enhancement in orthogonal and opposite directions relative to the upcoming target motion.

To further examine differences between the two subpopulations, we repeated the optimal linear model analysis from *Figures 5* and *6* separately on each subpopulation. Linear models generated from the two subpopulations perform roughly the same in predicting eye velocity during the single direction block (*Figure 9A,D*). Subpopulation 1 predicts eye movement during preparation in the same direction as the upcoming visual motion and eye movement (*Figure 9B,C*). Subpopulation 2 predicts no significant eye movement during preparation (*Figure 9E,F*). However, it is worth pointing out that the spread around zero for horizontal and vertical eye velocities is quite large. These features are consistent with Subpopulation 2 dialing up visual-motor gain in a directionally-nonspecific manner and they suggest that the nonspecific component of visual-motor gain is fairly uniform across directions.

In our system, we not only characterize the population activity during preparation and movement, but also understand why it reorganizes in the way that it does and how that reorganization improves the performance of pursuit eye movements. A full analysis of FEF$_{SEM}$ neural responses during pursuit eye movements shows that (1) there is a partial reorganization of the population between preparation and movement, (2) the reorganization is not complete enough to place the population activity in a movement-null dimension during preparation, (3) output control of the gain of visual-motor transmission rather than of movement kinematics allows the reorganization to be incomplete without causing movement, and (4) the existence of two subpopulations of neurons explains the partial

reorganization of population activity and allows the behavior to be preferentially prepared for target motion in an expected direction with a lower level of readiness for all directions of motion.

## Discussion

### Preparatory activity without movement

Preparatory activity is ubiquitous in motor systems. In advance of all but the most short-latency and reflexive movements, preparatory activity appears at least in the primary motor cortex (*Tanji and Evarts, 1976*), the premotor cortex (*Weinrich and Wise, 1982*), the frontal eye fields for both saccades and pursuit (*Bruce and Goldberg, 1985*; *Mahaffy and Krauzlis, 2011*; *Tanaka and Fukushima, 1998*), the parietal cortex (*Mazzoni et al., 1996*; *Shadlen and Newsome, 2001*), the cerebellum (*Gao et al., 2018*), the superior colliculus (*Dorris et al., 1997*; *Glimcher and Sparks, 1992*; *Munoz and Wurtz, 1995*; *Wurtz and Goldberg, 1972*), and even in last-order interneurons in the spinal cord (*Prut and Fetz, 1999*). It seems clear that preparatory activity plays a crucial role in setting the state of the nervous system leading up to movements. Many of the neurons that show preparatory changes in firing rate seem to play key roles in selecting, initiating, and/or guiding movement. Yet, preparatory activity evolves without actually causing movement. Why?

Different mechanistic explanations, all supported by data, have been proposed for different motor systems. The earliest solution, proposed for saccadic eye movements, is a 'gate' or 'switch' (*Keller, 1974*). The neural circuit that implements the switch is in the brainstem (*Cohen and Henn, 1972*; *Evinger et al., 1982*; *Luschei and Fuchs, 1972*). It has known internal connectivity, documented by electrophysiology and anatomical tracing with horseradish peroxidase (*Hikosaka et al., 1978*; *Langer and Kaneko, 1990*). The connectivity leads to switching behavior in computational models and the output pathways from the brain's switching circuit are appropriate to prevent saccades during preparation until it is time for the movement (*Fuchs et al., 1985*; *Robinson, 1973*).

A more recent solution, proposed for arm movements, suggests that the population of neurons in the motor cortex operates as a dynamical system (*Churchland et al., 2010*; *Churchland et al., 2012*). During the preparation of a movement, the population activity evolves within a set of dimensions that are 'movement-null' in the sense that population output is orthogonal to a later set of states that exists in 'movement-potent' dimensions (*Elsayed et al., 2016*; *Kaufman et al., 2014*). Analysis of population activity in the motor and pre-motor cortex supports the dynamical system explanation.

A third mechanism, based on our and others' research in smooth pursuit eye movements, involves the use of motor cortical output signals not to drive movement, but instead to modulate the strength of sensory-motor transmission. In the pursuit system, several lines of evidence, most notably the effects of microstimulation, imply that the output from the FEF$_{SEM}$ modulates the gain of visual-motor transmission (*Nuding et al., 2009*; *Tanaka and Lisberger, 2001*).

We imagine that all these mechanisms could be used in different blends by different motor systems. So, the challenge is to identify the mechanisms that prevent preparatory activity from causing movement in as many different movements as possible, explore the blend of mechanisms deployed by each motor system, and find the neural substrates of the different mechanisms.

### The use of FEF$_{SEM}$ output as a gain signal allows preparation to progress without causing movement

In the present paper, we present evidence that the smooth pursuit eye movement system emphasizes gain-modulation as a mechanism to allow preparation to progress without causing movement. Even though preparatory activity in FEF$_{SEM}$ is movement-potent in the sense that it shares dimensions with pursuit-related activity of the same population, preparatory activity fails to cause smooth eye movement because its role is to modulate the strength of movement-potent visual motion signals that are not present during preparation. Without the visual motion signal that it acts upon, gain can be dialed up without causing movement. The smooth eye movement output of the system will remain zero.

In contrast, our population analysis fails to support the dynamical systems explanation of orthogonal preparatory- and movement-related neural dimensions. The preparatory- and pursuit-related neural dimensions are neither completely orthogonal nor completely overlapping. Thus, there is a

reorganization of the population activity in $FEF_{SEM}$ between preparation and pursuit, but the reorganization is not complete enough to push preparatory activity into a movement-null dimension. In our system, we are able to understand in terms of the responses of its constituent neurons why the population reorganizes partially rather than fully, and we have a functional explanation for why the system might operate in this regime. We identified two subpopulations of cells in our data and one of them ('Subpopulation 2') seems to be responsible for the partial reorganization of neural dimensions between preparation and pursuit (*Figure 4D*). Because the reorganization is not sufficient to prevent movement during preparation, we propose that Subpopulation 1 creates a highly-directional enhancement of visual-motor gain while Subpopulation 2 is responsible for a directionally non-specific component of preparatory visual-motor gain enhancement, in effect providing a rigid and fairly uniform prior for the direction of target motion. A potential benefit of the regime created by the two Subpopulations is that the system will be fully ready for an expected direction of target motion and never be caught entirely flat-footed if target motion occurs in a completely unexpected direction.

Previous work suggests that gating mechanisms play minor roles for pursuit. The discharge of the neurons involved in the switch for saccades is incompatible with the idea that pursuit might use the same switch circuit (*Missal and Keller, 2002*). The firing of omnipause neurons pauses completely during the execution of saccades of any direction and amplitude but exhibits directionally-tuned decrements of firing rate that are graded according to eye speed during smooth pursuit eye movement. While one could argue that the use of an absolute versus a dynamic gate to prevent activity from causing movement is a matter of semantics, we are suggesting something fundamentally different in this paper: movement is not prevented during modulation of $FEF_{SEM}$ activity by the action of an inhibitory gate, rather $FEF_{SEM}$ output is controlling the gate. Indeed, it is possible that $FEF_{SEM}$ output interacts with omnipause neurons. Omnipause neurons are anatomically connected to at least the saccadic frontal eye fields and their responses during pursuit align closely with properties of visual-motor gain (*Langer and Kaneko, 1990*; *Missal and Keller, 2002*; *Schwartz and Lisberger, 1994*).

Our findings are not due to the trivial conclusion of a lack of importance of $FEF_{SEM}$ in the generation of pursuit eye movement. Lesioning and inactivating $FEF_{SEM}$ results in large decrements of pursuit speed and an increase in the number of catch-up saccades (*Keating, 1991*; *Shi et al., 1998*). The output of $FEF_{SEM}$ (and possibly partner areas such as the Supplementary Eye Fields *Missal and Heinen, 2001*) is necessary to allow visual motion signals to drive the oculomotor machinery (*Tanaka and Lisberger, 2001*). Furthermore, there are trial-by-trial correlations between $FEF_{SEM}$ activity during pursuit and eye velocity (*Schoppik et al., 2008*). The fact that microstimulation of $FEF_{SEM}$ causes smooth eye movement seems to contradict our conclusion that the output from $FEF_{SEM}$ is modulatory. We suspect that the artificial situation created by micro-stimulation will drive activity in brainstem oculomotor neurons simply because $FEF_{SEM}$ is anatomically connected to the motor system. Beyond general anatomical connectivity, micro-stimulation-generated smooth eye movements may say little about how $FEF_{SEM}$ output is actually used downstream.

## Comparative advantages of different mechanisms for preventing movement during preparation

We suggest that the brain uses multiple mechanisms to prevent preparatory activity from causing unwanted and premature movement, and that the primary mechanism used by any particular movement system will depend on the organization of that particular system. Further, a system could use multiple mechanisms at the same time, perhaps emphasizing one or another.

The short latency and sensory-driven features of pursuit may mitigate in favor of a system where preparatory activity is used to modulate sensory-motor transmission. In pursuit, movement initiation is driven directly, with relatively short latencies, by sensory inputs from extrastriate visual area MT (*Komatsu and Wurtz, 1989*; *Newsome et al., 1988*). Outputs from MT and $FEF_{SEM}$ converge downstream in the pursuit circuit to control the activity of motoneurons (*Lisberger, 2010*; *Mustari et al., 2009*). Importantly, preparatory modulation of visual-motor gain is a genuine preparatory signal. $FEF_{SEM}$ preparatory responses encode the expected speed and direction of the upcoming target motion and $FEF_{SEM}$ seems to set the state of the pursuit system during preparation in a way that sculpts the subsequent visual motion signals (*Darlington et al., 2018*).

In the arm movement system and the motor cortex, it may be difficult to prevent activity in the motor cortex from affecting the firing of motoneurons, and so it is ideal to nullify the preparatory activity by contriving for it to reside in a movement-null dimension. Here, the link from sensory input to motor output is less direct than for pursuit and the output from the primary motor cortex and the premotor cortex affect the firing of motoneurons directly, over monosynaptic and disynaptic pathways (*Dum and Strick, 1991*; *Griffin et al., 2015*; *Kuypers and Brinkman, 1970*). That said, it is possible that gain control mechanisms also operate in arm movements, both to implement the essential function of adjusting to different contexts, and to allow preparatory activity without movement. Just as we think that FEF$_{SEM}$ is responsible for the preparatory aspects of pursuit and area MT is responsible for the movement drive, recent work in the mouse motor system revealed subpopulations of motor cortex neurons with distinct roles in preparation and movement (*Economo et al., 2018*). Furthermore, last-order interneurons in the spinal cord show arm-movement-related preparatory activity that could reflect motor cortical modulation of the gain of spinal reflex circuits (*Prut and Fetz, 1999*).

In saccadic eye movements, the need for high degrees of accuracy and precision in a totally open-loop movement probably requires the preparatory activity that exists at least in the frontal eye fields, the superior colliculus, and the parietal cortex, but also requires an absolute brake on movement until all the brain's ducks are in a row (*Glimcher and Sparks, 1992*; *Hanes and Schall, 1996*; *Mazzoni et al., 1996*). Here, the switching circuit in the brainstem prevents any movement until the activity that guides the direction and amplitude of the impending saccade is fully prepared to make a movement that is at the same time really quick and accurate (*Robinson, 1973*). For example, a monkey executes a 20 deg saccade in 40 ms with accuracy of better than 5% (*Fuchs, 1967*). By contrast, pursuit eye movements are much slower and less accurate, and arm movements rely partly on feedback to achieve maximum accuracy.

Preparatory activity is ubiquitous, but the different features of different motor systems may require different mechanisms to prevent movement during preparation. A switching circuit could exist for movements other than saccades, perhaps in a softer, more compliant form. Gain control, may be an ideal way to establish context sensitivity and adjust the responsiveness to descending signals in systems with short latencies and graded responses. The dynamical system regime may be specialized for neural systems where the neurons involved in preparation make synaptic connections directly onto final motor pathways.

## Materials and methods

### Experimental model and subject details

As described in *Darlington et al. (2018)*, two male rhesus monkeys aged 8–10 years and weighing between 12 and 14 kg were used in the neurophysiology experiments (*Darlington et al., 2018*). The same two monkeys were used for the new behavioral experiments presented here. The monkeys underwent surgeries to implant head-restraining hardware onto the skull and a scleral search coil to track eye movements. Following recovery, monkeys were trained to fixate and smoothly track moving visual targets. During experiments, analog signals produced by the scleral search system recorded vertical and horizontal eye position. Eye velocity-related signals were obtained by passing the position-related signals through an analog circuit that differentiated signals from DC to 25 Hz and rejected signals at higher frequencies (−20 dB/decade). Eye position and velocity signals were sampled at 1 kHz and stored for offline data analysis. After training, monkeys underwent another surgery in which a craniotomy was performed in an area centered over the smooth eye movement region of the frontal eye fields (FEF$_{SEM}$). A sealable, titanium chamber was fixed to the skull over the craniotomy. All procedures received prior approval by Duke's *Institutional Animal Care and Use Committee* and complied with the National Institutes of Health's *Guide for the Care and Use of Laboratory Animals*.

### Method details
#### Neurophysiological experiments

We present new analyses of neural data that were collected using methods detailed in *Darlington et al. (2018)*. Briefly, we made extracellular single unit recordings in awake, behaving

monkeys. Three *Thomas Recording* tetrodes were introduced into FEF$_{SEM}$ for isolation of single units. Visual stimuli were displayed on a 24-inch gamma-corrected CRT monitor with a refresh rate of 80 Hz placed 40 centimeters from the monkeys' eyes, creating a field of view that spanned 62 (horizontal) by 42 (vertical) degrees.

Each experiment was divided into discrete trials. All trials began with the monkey fixating a central black spot for 800–1600 ms, after which the fixation target was replaced with a pursuit target. The pursuit target was either a patch of 72 dots within a 4-degree aperture or a Gabor function see *Darlington et al. (2018)*. The pursuit target initially underwent 100 ms of coherent local pattern motion at a defined speed and direction within a stationary, invisible aperture. The target and aperture subsequently moved *en bloc* with the same parameters as the previous local motion. The initial use of local motion followed by global motion helps avoid early catch-up saccades and has similar advantages as the Rashbass 'step-ramp' used in pursuit paradigms for spot-targets (*Rashbass, 1961*). Monkeys were rewarded with juice for successfully tracking the center of the target within a 4 × 4 degree window.

During neurophysiological recording, we ran two different behavioral experiments. In the '8-direction' experiment, targets moved at 15 deg/s in a direction randomly chosen from the four cardinal and four oblique directions. In the 'single direction' experiment, targets moved in only one direction (chosen from the four cardinal and oblique directions). Thus, target direction was perfectly predictable. The 'single-direction' task was split up into blocks of trials that provided visual motion of different blends of speed. In the fast-context, 80% of the trials moved at 20 deg/s and 20% of the trials moved at 10 deg/s. In the slow-context, 80% of the trials moved at 2 deg/s and 20% of the trials moved at 10 deg/s. In a control-context, 100% of the trials moved at 10 deg/s. In both of the experiments, pursuit targets of high- (100%) and low-contrast (6 or 12%) were interleaved randomly.

## Behavioral assessment of gain of visual-motor transmission

We performed two separate sets of behavioral experiments, without neurophysiological recordings, as a modified version of the single-direction experiment described above. We presented the same blends of targets, directions, and speeds but the pursuit target was present through the entire trial, including during fixation before the onset of target motion. On approximately half of the trials, the pursuit target delivered a brief pulse that comprised a 5 deg/s step of target speed for 50 ms. Pulses occurred at a random time during fixation, 550–1350 ms after fixation onset. In the speed context experiment, pulses provided motion in the same direction as the impending target motion. In an experiment that tested direction selectivity, pulses provided motion in one of four directions, the same direction, opposite direction, or one of the two directions orthogonal to the direction of target motion in the single-direction experiment. The purpose of the pulse in target velocity was to probe the state of visual-motor gain in the pursuit system. We confirmed that the brief pulse of target velocity evokes responses with all the same features discovered before with sinusoidal perturbations (data not shown), and we chose to use the pulse to better assess directional specificity of gain modulation during fixation (*Schwartz and Lisberger, 1994*).

## Comparison with arm movement paradigm

Our behavioral paradigm for defining movement potent dimensions was very similar to the behavioral paradigm used in *Kaufman et al. (2014)*. The random 8-direction pursuit task is the eye movement analogue to the center out reaching task. One notable difference is that we used a single-direction task to probe the preparatory activity whereas Kaufman et al. used the preparatory epoch during the center out task. We did not use explicit cues during the 8-direction block to allow for preparation to occur. Instead we used the single-direction block to allow for expectation-driven preparation. Several benefits to our approach include the use of single-direction eye movement epoch data for cross validation (*Figure 5D*) and prevention of confounding variables like visual cues. There were no differences in the methods by which we collected neurophysiological data. Most neurons were recorded during different experimental sessions. This is consistent with half of the data presented in *Kaufman et al. (2014)* and all of the data presented in *Elsayed et al. (2016)*.

## Quantification

### Demixed principal component analysis

The method of demixed principal component analysis (dPCA) is described in detail by Koback et al. (2016). Briefly, dPCA is a semi-supervised dimensionality reduction technique that in addition to capturing the majority of the variance present in the data (e.g. PCA) pulls out, or 'demixes', features of the data that are task-dependent (i.e. stimulus, decision, etc.). It does so by first decomposing a neural matrix, $X$, with n rows that is mean-centered on zero into a sum of averages taken over various task parameters, $\phi$.

$$X = \sum_{\phi} X_{\phi} + X_{noise} \tag{1}$$

The core of dPCA is to then find a low dimensional encoding/decoding model (FD) for *each $X_{\phi}$* that minimizes the loss function:

$$L = \sum_{\phi} \frac{1}{2} \left\| X_{\phi} - F_{\phi} D_{\phi} X \right\|^2 \tag{2}$$

FD produces an n x n matrix A, which could normally be solved via classical regression, yielding the ordinary least squares solution. However, A is not full rank and is instead solved as a reduced-rank regression problem via singular value decomposition.

We used the MatLab code publicly provided from the Machen's lab to perform the dPCA presented in *Figure 4* (https://github.com/machenslab/dPCA). Our goal in using dPCA was to discover features of the FEF$_{SEM}$ population response during preparation that are related to the expectation of upcoming direction of visual motion and eye movement. As direction is not predictable during the eight-direction block, we relied on data from the single-direction task to probe reliable preparatory responses. In this way, expectation of direction is implicit and built from experience as monkeys tracked targets in the same direction typically for about 700–800 consecutive trials. The direction of the single-direction task varied across experiments and was chosen at various points relative to the preferred direction of the neurons under study. In order to get a sense for the neural representation of expected upcoming target direction and to pool as much of our data together, we subsampled our population of FEF$_{SEM}$ neurons, limiting ourselves to neurons where we ran multiple single-direction experiments with target motion both toward and away from their preferred directions. Toward the preferred direction was considered to be any single-direction experiment run within 90 degrees of the preferred direction and away from the preferred direction was considered to be any experiment run greater than 90 degrees away from the preferred direction. Preferred directions were calculated from pursuit responses of each neuron to trials in a direction tuning block of trials (see *Statistics* section). Thus, the sole condition used in our dPCA is the expected (and impending) direction of target motion relative to the preferred direction.

Of the 146 neurons that we recorded, we were able to collect single-direction experiments run both toward and away from the preferred direction in 63 of our neurons. We used 1400 ms of firing rate during fixation for each of the two conditions (towards and away from preferred direction). Therefore, the $X$ in our analysis was a 63 x 2800 matrix.

### Linear model relating FEF$_{SEM}$ to eye velocity

We analyzed neurophysiological data and eye velocity from the random 8-direction block of trials to generate a linear model relating the modulation of firing rate of our population of FEF$_{SEM}$ neurons to changes in eye velocity:

$$E = W \times N_{8D} \tag{3}$$

Here, $E$ is a 2 × 2000 matrix of eye velocity (horizontal, vertical velocity x eight directions, 250 ms of data per direction) averaged across all experiments. $N_{8D}$ is a 297 × 2000 matrix of firing rate modulation during the random 8-direction experiment (297 single-direction neural recordings from 146 neurons x eight directions, 250 ms of data per direction). $W$ is the 2 × 297 wt matrix that relates the firing rate of each neuron to modulation of horizontal and vertical eye velocity.

We took several steps to combine neural data collected during different experiments. The single-direction experiments were run in different directions for different cells in an attempt to sample the full tuning curve of our population. Because we were interested in how modulation of our population of cells predicts changes of eye velocity relative to the single direction, we rotated the tuning curves of each cell so that they were aligned according to the direction of the single-direction experiment. For example, if the single-direction experiment was run at 90 degrees, then 90 degrees during the random 8-direction experiment was considered 0 degrees, 180 degrees was considered to be 90 degrees, 270 degrees was considered to be 180 degrees, and 0 degrees was considered to be 270 degrees. We rotated eye velocity in the same way and averaged across experiments to get the 2 × 2000 $E$ matrix. For each neuron, firing rate responses to motion in each of the eight directions were averaged across trials, smoothed with a Gaussian filter (σ = 20 ms, similar results were obtained using σ = 10 ms), subtracted to have a mean of zero across all conditions, and normalized to the range across all conditions. To accommodate known lags between firing rate and movement, behavioral data was shifted forward in time by 25 ms. We repeated the analysis with shifting by 50 and 0 ms, but 25 ms yielded the best cross validation shown in *Figure 5D*. Nevertheless, repeating the analysis with the other temporal shifts yielded similar conclusions (data not shown). The weight matrix, $W$, was solved using regularized (ridge) regression.

Firing rates during fixation from the single-direction paradigm were processed in a similar manner. For each neuron, we averaged firing rate across trials, aligned to fixation onset. Each trial contributed data up to the time of the onset of target motion in that trial. We discarded the first 150 ms of averaged data to avoid any effects from the beginning of the trial and/or the saccade back to the center of the screen from the previous trial. We also discarded the last 50 ms of averaged data since there were too few trials with long enough fixation durations to generate a meaningful average. Again, firing rates were subtracted to have a value of zero at the start of the trial and normalized using the same ranges that were used for the normalization of the random 8-direction firing rates. The firing rates during fixation then were passed through the linear model generated from solving $W$ of *Equation (1)* to obtain $PE$, the eye velocity predicted from preparatory modulation of firing rate:

$$PE = W \times N_{1D} \tag{4}$$

Here, $N_{1D}$ is a 297 × 1400 matrix of firing rates (297 data points from 146 neurons x 1400 ms of fixation data; firing rates were trial-averaged, Gaussian-smoothed, σ = 20 ms, similar results with σ = 10 ms) and $PE$ is a 2 × 1400 matrix of predicted eye velocity (horizontal, vertical velocity x 1400 ms of fixation). Predicted speeds (PS) were computed using standard trigonometry based on the horizontal and vertical components of the predicted eye velocity.

For the single trial analysis performed in *Figure 6C and D*, we used the same $W$ matrix obtained from *Equation (1)*. However, we computed preparatory firing rate modulation on single trials for each neuron as the average firing rate during the last 100 ms of fixation minus the average firing rate 150–250 ms after fixation-onset. Trials were matched by fixation time across cells and then randomly sampled (one trial per cell). These single trial data were normalized to the same range used to normalize $N_{8D}$ and $N_{1D}$ and passed through the linear model. For each of 1000 repetitions of the analysis, we computed the predicted eye direction as the arctangent of the predicted vertical and horizontal components of eye velocity and predicted eye speed as the square root of the sum of the squares of the predicted horizontal and vertical components.

## Principal component analysis

We performed dimensionality reduction independently on preparatory- and pursuit-related firing rates using Principal Component Analysis (PCA) on their respective data matrices ($N_{prep}$ and $N_{purs}$). $N_{prep}$ is a 297 neuron x 1400 ms of preparatory-related firing rate matrix and $N_{purs}$ is a 297 neuron x 750 ms of pursuit-related firing rate matrix (125 ms x 2 contrasts x two target speeds). All firing rates were trial-averaged and Gaussian smoothed (σ = 20 ms, similar results were obtained with σ = 10 ms). As is standard for PCA, we preprocessed each data matrix by normalizing rows (firing rates for each neuron) to have unity range and then centering firing rates on their means by subtracting off the row means. Each PC represents an orthogonal vector in N = 297 (or N = 160 for the bootstrap analyses) dimensional neural space. We chose to focus all of the subsequent analysis on the first

identified PC for preparation and movement (PC$_{Prep1}$ and PC$_{Purs1}$), because they each captured at least 70% of the variance of the full responses (*Elsayed et al., 2016*). The angle between PC$_{Prep1}$ and PC$_{Purs1}$ was calculated as:

$$\theta = arccosine\left(\frac{PC_{Prep1} \cdot PC_{Purs1}}{|PC_{Prep1}| \times |PC_{Purs1}|}\right) \quad (5)$$

We also determined how the PCA component loadings for each unit changed between PC$_{Prep1}$ and PC$_{Purs1}$ by calculating the ratio PC$_{Prep1}$/PC$_{Purs1}$.

## Pulse analysis

Trials with saccades, detected with a combination of acceleration and velocity thresholds, in the window from pulse onset to 200 ms after visual motion onset were discarded before averaging the eye speed responses to pulses in matched conditions. The magnitude of the eye speed response was estimated as the maximum eye speed achieved across the 200 ms following pulse onset.

## Statistics

We did not use formal tests to predetermine sample size, but our sample sizes are comparable to those reported in previous studies. Data were collected from two rhesus monkeys and consist of 321 data points collected from 164 neurons across 95 experimental sessions. We used only cells for which at least four trials/direction were collected during a direction-tuning block. This brought the sample size to 297 data points from 146 neurons: 80 neurons (187 data points) from monkey RE and 66 neurons (110 data points) from monkey XT. For the behavioral pulse experiments, 26 experiments were run in two monkeys for the speed context analysis (*Figure 3*) and 30 experiments were run in two monkeys for the effect of pulse direction experiments (*Figure 4*). Of the 30 pulse direction experiments, 27 had an early pulse onset time point of 400 ms for the same direction (*Figure 2*). Hypothesis testing was accomplished using a mix of two-sided bootstrap statistics, two-sided Wilcoxon signed rank tests, and two-sided Wilcoxon rank sum tests. Parametric hypothesis testing led to the same conclusions. The bootstrap procedure used in *Figures 7* and *8* and Supplementary Figure 1 randomly sampled 20 units from each of the eight groups with different preferred directions relative to the direction of the experiment. We randomly sampled without replacement to avoid rank issues when performing the linear fitting and PCA. We chose 20 units because the smallest number of units in one of the 8 groups of preferred directions is 24. Preferred direction was calculated for each neuron as the weighted circular mean of the responses during the 8-direction block. This gives the mean parameter of a von Mises distribution fit to the direction tuning data. Our bootstrap approach was chosen to generate populations with a uniform distribution of preferred directions as a way of controlling for any biases of preferred direction in our data collection. The bootstrap was repeated for 1000 iterations.

## Acknowledgements

We thank Stefanie Tokiyama, Bonnie Bowell, Scott Ruffner, and Steven Happel for technical assistance, and the other members of our laboratory for helpful comments on the manuscript. We thank J Patrick Mayo for providing the figure panel found in our response to the reviews. Research supported by R01-EY027373 (SGL) and F30-EY027684 (TRD).

## Additional information

### Funding

| Funder | Grant reference number | Author |
| --- | --- | --- |
| National Institutes of Health | R01-EY027373 | Stephen G Lisberger |
| National Institutes of Health | F30-EY027684 | Timothy R Darlington |

The funders had no role in study design, data collection and interpretation, or the decision to submit the work for publication.

## Author contributions
Timothy R Darlington, Conceptualization, Formal analysis, Investigation, Methodology, Writing - original draft, Writing - review and editing; Stephen G Lisberger, Conceptualization, Supervision, Funding acquisition, Methodology, Writing - review and editing

## Author ORCIDs
Timothy R Darlington https://orcid.org/0000-0001-5534-7552
Stephen G Lisberger http://orcid.org/0000-0001-7859-4361

## Ethics
Animal experimentation: All procedures received prior approval by Duke's Institutional Animal Care and Use Committee (protocol A085-18-04) and were in compliance with the National Institutes of Health's Guide for the Care and Use of Laboratory Animals.

## Decision letter and Author response
Decision letter https://doi.org/10.7554/eLife.50962.sa1
Author response https://doi.org/10.7554/eLife.50962.sa2

## Additional files
### Supplementary files
• Transparent reporting form

### Data availability
Source data have been provided for each figure.

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
