## [Decision Letter]

**Acceptance summary:**

The study investigates how activity in a specific region of the frontal eye field changes when smooth pursuit eye movements switch from preparation to execution. They propose a novel framework based on visuomotor gain to account for the behavioural and neural data collected from monkeys, different from the movement potent framework that has been previously proposed for the skeletomotor system.

**Decision letter after peer review:**

Thank you for submitting your article entitled "Mechanisms that allow cortical preparatory activity without inappropriate movement" for peer review at *eLife*. Your article is being evaluated by three peer reviewers, and the evaluation is being overseen by a Reviewing Editor and Joshua Gold as the Senior Editor.

After consultation with each other and with the Reviewing Editor and Senior Editor, the reviewers have identified a list of essential revisions, involving extensive new analyses, which are listed below.

The present study aims to understand whether pseudo-population activity of FEFsem neurons is better explained by a visual-motor gain hypothesis or by the movement-potent/null space framework for the skeletomotor system. Using a brief pulse of visual motion during the preparatory period to produce transient eye motion during fixation, the authors found that the size of eye motion is modulated by factors such as the timing of pulse within the fixation period, direction of pulse motion, and expectation of target speed. This result was interpreted as matching the predictions of the visuomotor gain hypothesis. The authors further found that: 1) an optimal linear model, parameterized by weight estimates from one smooth pursuit task with 8 directions, was used on the same FEFsem neuron population during the preparatory period, and the model predictions appeared to match the actual eye velocity during fixation when there wasn't any; and 2) the principal components (PCs) of the population response measured during preparatory and movement initiation phases were separated by about 65 deg, as opposed to 90 deg as predicted by the movement-null space framework. These results were interpreted as evidence against the movement-null space framework.

All three reviewers (including the reviewing editor) find this paper interesting, well-written, and timely for the field. However, they converged on a common set of concerns about the data analysis.

1) The evidence against the movement-null space frame is indirect. For example, the approach is to find the preparatory subspace in one task and look at what happens in another task. Extensive additional analyses are needed to make the current work comparable to the studies (e.g., Kaufman et al.).

2) Some data analyses are confusing, such as artificially rotating the tuning of some neurons to build a neural space that sometimes includes the same neuron on different axes. These analyses should be either simplified or more clearly explained and justified.

3) The PCA analysis should be improved; e.g., using dPCA.

4) It would be meaningful and necessary to applied subspace analysis to the data from the experiment that provided the behavioral evidence (e.g., Figure 2).

Reviewer #1:

In this manuscript Darlington and Lisberger address the question how preparatory activity in the smooth pursuit eye movement system is prevented from actually driving eye movements. Different solutions have been proposed for different motor systems. The saccadic eye movement system has a gate (omnipause neurons) preventing preparatory activity from activating burst neurons. An analysis of primary motor and premotor activity in the arm movement system has shown that preparatory population activity avoids state space dimensions that result in driving motor output. For the smooth pursuit eye movement system, the authors propose that preparatory activity in the frontal eye field (FEF) controls the gain of a sensorimotor transformation process that is actually driven by visual motion information. An analysis of the FEF population activity indicated that the preparatory activity does not completely avoid the dimensions related to motor output, such that the preparatory activity would be expected to cause eye movements if the neural activity in FEF were responsible for driving smooth pursuit.

1) One of the key elements supporting the authors' claims is a comparison of the time course of preparatory activity in FEF with the time course of the behavioral effect of brief motion pulses. Ideally, these data would have been collected in a combined experiment. For this manuscript they were collected in separate experiments. This is not necessarily problematic as they were collected from the same monkeys. However, both time courses are likely affected by the animals' temporal expectations about upcoming eye movements. The timing of the experiments (distribution of time intervals between fixation onset and onset of the continuous motion cue driving pursuit, not the pulses) should therefore have been identical to be able to make the assumption that the animals' expectation was probably similar. The manuscript is not explicit about whether this was the case. Was it?

Other than that, I don't have any major issues. The data analysis is solid, and the manuscript is very clearly written.

Reviewer #2:

Darlington and Lisberger present a well-written and interesting study that explores how activity in a specific region of the frontal eye field (FEFsem) transitions between preparation and execution of smooth pursuit eye movements. The overall objective is to understand whether pseudo-population activity of FEFsem neurons is better explained by the visual-motor gain hypothesis, originally developed by Lisberger and colleagues, or by the movement-potent/null space framework championed recently by Shenoy and colleagues for the skeletomotor system. The authors provide three results that, in their opinion, support the visual-motor gain hypothesis and refute the movement-null space perspective. I will summarize the three findings and then evaluate the authors' interpretations.

1) Injecting a brief pulse of visual motion during the preparatory period produces transient eye motion during fixation. The size of eye motion depends on several factors (when during the fixation period, direction of pulse motion, and expectation of target speed). The authors view the results as consistent with and build logically on the visual-motor gain hypothesis as a mechanism for movement initiation, particularly smooth pursuit. In my view, their interpretation is reasonable.

2) As a test of the movement-potent space hypothesis, the authors estimate the contribution of each neuron/condition to the initiation of smooth pursuit during one task condition (8 directions randomly interleaved). They then use these weights on the preparatory activity of the same population during a different task (repeat trials of same direction) and find that the model predicts changes in eye velocity (during fixation) when there isn't any. This finding is interpreted as evidence against the movement-potent space framework.

In my view, the analysis is not rigorous enough to reach this conclusion. Is it safe to assume, as the authors seem to do, that the population response for movement initiation during the two tasks is the same? How well do the weights for the 8D task predict pursuit initiation for the 1D task? Also, how well do weights obtained from 1D task during pursuit predict eye velocity during preparation in the same 1D task? These factors must be addressed before appreciating the importance of the result emphasized in the manuscript. Also see dPCA comment below.

3) In a related analysis, the authors evaluated the principal components (PCs) of the population response measured during preparatory and movement initiation phases. They found that the 1st PC for each period accounts for ~70% of the variance and that the mean angle between these PCs is about 65 deg, which is not equal to the 90 deg required if preparatory period activity is in a movement-null space. In addition, they identified a subpopulation of neurons that accounts for the observed change in angle between the PCs.

Given that the angle of separation between the primary preparatory and movement PCs is not 90 deg, the authors de-emphasize the potential importance of the movement potent/null space framework and instead view favorably the visual-motor gain hypothesis. However, they do state, "two mechanisms for preventing movements might be better than one". My view on their data is that the population FEFsem definitely resides in different subspaces during preparatory and movement initiation periods. In other words, if we relax a strict interpretation of orthogonality of movement and null subspaces, then these data are very much in line with potent/null space framework. This very much contrasts the take-home message the authors deliver.

Moreover, it is not clear that PC analysis is the best/correct analysis to use for the current data. The authors should consider using demixed principal component analysis (dPCA; https://elifesciences.org/articles/10989; https://www.eneuro.org/content/3/4/ENEURO.0085-16.2016.long). An interesting outcome of previous studies using dPCA is that the primary PCA is generally condition independent and could reflect the large overlap in angle between the PCs. A separation is more appreciable with secondary and tertiary PCs. The authors should pursue this angle of inquiry.

Finally, there is a glaring omission in the analysis. The first part of the paper shows a robust behavioral result, in which a brief visual motion perturbation produces an eye movement during a period typically associated with movement preparation. What subspace is spanned by FEFsem population during the transient eye movement response? Does the activity remain aligned with preparatory PCs (evidence against functional importance of movement-null space) or does it shift to movement PCs (evidence to support the movement-potent hypothesis)? It seems to me that this analysis is crucial to understand population activity control of smooth pursuit by FEFsem.

In summary, a study of how population activity relates to movement preparation and generation is timely. Tests of whether new hypotheses developed in skeletomotor systems generalize to the oculomotor system is very much needed, and this study takes a step in the direction. However, the analysis isn't rigorous enough to reach a reliable conclusion.

Reviewer #3:

Darlington and Lisberger present a previously undescribed "mechanism" by which the smooth eye movement (SEM) region of FEF could prepare pursuit movements without causing them -by dialing up the visuomotor gain. Moreover, they provide evidence in the SEM system against the "null subspace" hypothesis of preparation of arm movements. I find many aspects of the manuscript interesting but I have some concerns about their analytical approach

Comments:

1) Although I think that the potential differences in control of reaching movements and pursuits are interesting, I have concerns about how the authors tested the "null subspace" hypothesis. Overall, I think that the authors should replicate the methods proposed by Kaufman et al., rather than introduce a number of modifications whose implications are hard to grasp -at least for me. In more detail:

1a) All subspace analyses putatively capture activity that reflects some aspect of the underlying circuitry as well as the task (e.g, related to inputs to the neural population or outputs). However, the authors find the null subspace in a lower-dimensional version of the task that they probe (1-targets vs. 8-targets) that is also -I think- cued differently. My suggestion is that they do the analysis using the full 8-target data and considering the preparatory and pursuit epochs as Kaufman et al. did.

1b) Studies like Kaufman's assume that the relevant "neural information" is captured by activity patterns shared across the neural population; these patterns are probed by analyzing a state space in which axis is the activity of one recorded neuron. The authors adopted an approach that puzzled me: having multiple axes that reflect the same neuron after rotating its activity. I'd again suggest them to adopt the simpler approach of defining one axes per neuron, discarding neurons that were recorded multiple times during different conditions (at least I have trouble foreseeing all potential implications of their manipulations)

1c) I am again a bit confused about the potential effect of the authors rotating the tuning curves of each neuron in the analysis. Although I agree that there may be FEF neurons with these properties, I would avoid it because it implies assuming that the absolute coordinates of the target doesn't matter, or at least show that it doesn't influence the main result.

1d) Did the authors include all the recorded neurons in this analysis or only neurons that were significantly tuned to the task? Although studies in the motor cortices suggest that the population dynamics sampled from sufficiently large neural populations are the same irrespective of the sampled neurons (Trautman et al., 2018), I worry this may bias their results.

1e) Because of these concerns, I believe that the authors should repeat Figure 8 after calculating the preparatory and pursuit PCs as in Kaufman et al., 2014.

1f) Minor: I assume that the authors only compare the first pursuit PC and the first preparatory PC because they each capture a large fraction of the total neural variance. For a full assessment, I'd suggest an analysis like the one in Figure 4 of Elsayed et al., 2016, or the principal angle analysis used in Gallego et al., 2018.

2) The central idea of "gain control" is interesting, but I think the authors should pursue it further. For example, can single trial eye speed be predicted from neural activity? (I'd do this analysis for the data in Figure 2 and 3). Also, how do the authors think this gain leads to a transition from preparation to movement, by exceeding a threshold? I think some analyses and discussion in this regard are missing.

3) In the Results section the authors make the proposal that there may be "non-specific preparatory enhancement of visuomotor gain". I think they could address this directly by using demixed principal component analysis (dPCA) (Kobak et al., 2016). dPCA effectively performs semi-supervised dimensionality reduction and can be used to isolate population activity patterns that only depend on time (these would be the component the authors refer to) and components that depend on the direction of the movement (see, e.g., Kaufman et al., 2016, Gallego et al., 2018 for examples in motor cortex). Examining both the dynamics of these components (e.g., when do putative time-related and target-related components start ramping up?) and the weights (e.g., are there subpopulations of neurons?) could be interesting. Note that there's freely available code by Kobak et al.

4) Figure 2 shows strong effects, but it would be nice to see the data distributions, or at least box plot versions of (b) and (c). As to (d), would the authors see the same trend if they plotted single trial values or a 2D histogram? This figure focuses in neurons that had "positive preparatory firing rate modulation". What was their percentage among all recorded neurons? And among all modulated neurons? And what happens with these neurons? Same for the increase and decrease cells in Figure 3.

References:

GF Elsayed, AH Lara, MT Kaufman, MM Churchland and JP Cunningham. Reorganization between preparatory and movement population responses in motor cortex. Nature Communications 2016

JA. Gallego, MG. Perich, SN. Naufel, C Ethier, SA Solla and LE Miller. Cortical population activity within a preserved neural manifold underlies multiple motor behaviors. Nature Communications 2018

MT Kaufman, JS Seely, D Sussillo, SI Ryu, KV Shenoy and MM Churchland. The Largest Response Component in the Motor Cortex Reflects Movement Timing but Not Movement Type. eNeuro 2016

D Kobak, W Brendel, C Constantinidis, CE Feierstein, A Kepecs, ZF Mainen, X-L Qi, R Romo, N Uchida, C Machens. Demixed principal component analysis of neural population data. *eLife* 2016

MG Perich, JA Gallego, LE Miller. A Neural Population Mechanism for Rapid Learning. Neuron 2018

EM Trautmann, SD Stavisky, S Lahiri, KC Ames, MT Kaufman, DJ O'Shea, S Vyas, X Sun, SI Ryu, S Ganguli, KV Shenoy. Accurate estimation of neural population dynamics without spike sorting. Neuron 2018

[Editors' note: further revisions were suggested prior to acceptance, as described below.]

Thank you for resubmitting your work entitled "Mechanisms that allow cortical preparatory activity without inappropriate movement" for further consideration by *eLife*. Your revised article has been evaluated by Joshua Gold (Senior Editor), a Reviewing Editor, and two of the original reviewers.

The manuscript has been improved substantially. The addition of new analyses as well as the justification for why other requested analyses are not needed were satisfactory. Overall, the revised version is well written and of strong impact. It pushes back a bit against the increasingly dogmatic view of movement-potent and movement-null subspaces.

We have one remaining major concern that should be addressed:

1) It would be interesting to a) recompute weights relating FEFsem activity to eye velocity separately for subpopulations 1 and 2, and (b) predict eye velocities for each. Is there a difference in the predictive power for the two subpopulations? And how well do the results align with the proposed role of each subset (including for different directions)? A new figure can probably be committed to the results.

---

## [Author Response]

Reviewer #1:In this manuscript Darlington and Lisberger address the question how preparatory activity in the smooth pursuit eye movement system is prevented from actually driving eye movements. Different solutions have been proposed for different motor systems. The saccadic eye movement system has a gate (omnipause neurons) preventing preparatory activity from activating burst neurons. An analysis of primary motor and premotor activity in the arm movement system has shown that preparatory population activity avoids state space dimensions that result in driving motor output. For the smooth pursuit eye movement system, the authors propose that preparatory activity in the frontal eye field (FEF) controls the gain of a sensorimotor transformation process that is actually driven by visual motion information. An analysis of the FEF population activity indicated that the preparatory activity does not completely avoid the dimensions related to motor output, such that the preparatory activity would be expected to cause eye movements if the neural activity in FEF were responsible for driving smooth pursuit.1) One of the key elements supporting the authors' claims is a comparison of the time course of preparatory activity in FEF with the time course of the behavioral effect of brief motion pulses. Ideally, these data would have been collected in a combined experiment. For this manuscript they were collected in separate experiments. This is not necessarily problematic as they were collected from the same monkeys. However, both time courses are likely affected by the animals' temporal expectations about upcoming eye movements. The timing of the experiments (distribution of time intervals between fixation onset and onset of the continuous motion cue driving pursuit, not the pulses) should therefore have been identical to be able to make the assumption that the animals' expectation was probably similar. The manuscript is not explicit about whether this was the case. Was it?

We agree that this is a very important issue. Despite the current ordering of figures, the behavioral experiments were actually a follow-up to the neural analyses we performed on our FEFSEM data. We matched all experimental conditions, including timing, in order to maximize the validity of our comparisons between the neurophysiology and behavioral data sets. We have added a short paragraph in the relevant place in the Results to be more explicit about this.

Other than that, I don't have any major issues. The data analysis is solid, and the manuscript is very clearly written.

We’d like to thank the reviewer for the positive feedback. We are glad that the reviewer enjoyed our manuscript.

Reviewer #2:Darlington and Lisberger present a well-written and interesting study that explores how activity in a specific region of the frontal eye field (FEFsem) transitions between preparation and execution of smooth pursuit eye movements. The overall objective is to understand whether pseudo-population activity of FEFsem neurons is better explained by the visual-motor gain hypothesis, originally developed by Lisberger and colleagues, or by the movement-potent/null space framework championed recently by Shenoy and colleagues for the skeletomotor system. The authors provide three results that, in their opinion, support the visual-motor gain hypothesis and refute the movement-null space perspective. I will summarize the three findings and then evaluate the authors' interpretations.1) Injecting a brief pulse of visual motion during the preparatory period produces transient eye motion during fixation. The size of eye motion depends on several factors (when during the fixation period, direction of pulse motion, and expectation of target speed). The authors view the results as consistent with and build logically on the visual-motor gain hypothesis as a mechanism for movement initiation, particularly smooth pursuit. In my view, their interpretation is reasonable.2) As a test of the movement-potent space hypothesis, the authors estimate the contribution of each neuron/condition to the initiation of smooth pursuit during one task condition (8 directions randomly interleaved). They then use these weights on the preparatory activity of the same population during a different task (repeat trials of same direction) and find that the model predicts changes in eye velocity (during fixation) when there isn't any. This finding is interpreted as evidence against the movement-potent space framework.In my view, the analysis is not rigorous enough to reach this conclusion. Is it safe to assume, as the authors seem to do, that the population response for movement initiation during the two tasks is the same? How well do the weights for the 8D task predict pursuit initiation for the 1D task? Also, how well do weights obtained from 1D task during pursuit predict eye velocity during preparation in the same 1D task? These factors must be addressed before appreciating the importance of the result emphasized in the manuscript. Also see dPCA comment below.

We’d like to thank the reviewer for the insightful questions and we agree that it is important that they be answered. In Figure 5D, which was part of the original paper, we show that the weights calculated in the 8D task do an excellent job of predicting actual horizontal eye velocity for the 1D task. These same weights accurately predict 0 vertical eye velocity (i.e. accurately predict direction) for the 1D task; because this is a ridiculous looking figure with all points plotting at (0,0), we state this result explicitly in the text. We have now also included a supplementary analysis shown in Figure 6—figure supplement 3 to confirm that we obtained all the same results when the fitting and predicting were both done on the 1D task.

3) In a related analysis, the authors evaluated the principal components (PCs) of the population response measured during preparatory and movement initiation phases. They found that the 1st PC for each period accounts for ~70% of the variance and that the mean angle between these PCs is about 65 deg, which is not equal to the 90 deg required if preparatory period activity is in a movement-null space. In addition, they identified a subpopulation of neurons that accounts for the observed change in angle between the PCs.Given that the angle of separation between the primary preparatory and movement PCs is not 90 deg, the authors de-emphasize the potential importance of the movement potent/null space framework and instead view favorably the visual-motor gain hypothesis. However, they do state, "two mechanisms for preventing movements might be better than one". My view on their data is that the population FEFsem definitely resides in different subspaces during preparatory and movement initiation periods. In other words, if we relax a strict interpretation of orthogonality of movement and null subspaces, then these data are very much in line with potent/null space framework. This very much contrasts the take-home message the authors deliver.

We’d like to thank the reviewer for this feedback, which helped us to recognize a deficiency in our presentation in the original version of the paper. We have rewritten parts of the Results and Discussion to be transparent that while not orthogonal, there is clearly some reorganization between preparation and movement. We also added text to explain that the full set of data that we think argues against a simple and traditional “null-space” explanation for our findings. First, preparatory activity does evolve in a way that predicts movement, which is the strongest piece of evidence against the null/potent framework in the pursuit system. Second, we outline a broader (and in our view more compelling) function for this reorganization beyond simply preventing movement during preparation. Through an analysis that isolates the part of our population that is driving the partial reorganization, cells whose preferred direction was generally opposite the direction of the 1D experiment (that seemingly have a strong “prior” for that direction based on their preparatory responses), we connect this feature of the full population response to our behavioral finding of a directionally non-specific enhancement of visual motor gain (Figure 4A, orthogonal/opposite). We suggest that a more likely function of this reorganization is to allow the pursuit system to partially prepare for less likely directions of visual motion, so that it is never caught completely flat-footed in the event of unexpected target motion.

Moreover, it is not clear that PC analysis is the best/correct analysis to use for the current data. The authors should consider using demixed principal component analysis (dPCA; https://elifesciences.org/articles/10989; https://www.eneuro.org/content/3/4/ENEURO.0085-16.2016.long). An interesting outcome of previous studies using dPCA is that the primary PCA is generally condition independent and could reflect the large overlap in angle between the PCs. A separation is more appreciable with secondary and tertiary PCs. The authors should pursue this angle of inquiry.

We considered dPCA and read the paper cited by the reviewer, but decided in the end that our unbiased approach was actually more rigorous. We wanted to use an unbiased estimator of preparatory and pursuit dimensions to produce an unbiased result. Unless we have misunderstood the reviewer, we disagree that it would be a move rigorous approach to show that preparatory and pursuit secondary and tertiary PCs are different based on the results of semi-supervised dimensionality reduction, which defines preparation and pursuit as being different conditions.

Finally, there is a glaring omission in the analysis. The first part of the paper shows a robust behavioral result, in which a brief visual motion perturbation produces an eye movement during a period typically associated with movement preparation. What subspace is spanned by FEFsem population during the transient eye movement response? Does the activity remain aligned with preparatory PCs (evidence against functional importance of movement-null space) or does it shift to movement PCs (evidence to support the movement-potent hypothesis)? It seems to me that this analysis is crucial to understand population activity control of smooth pursuit by FEFsem.

Unfortunately, we do not have access to the data requested by the reviewer. The visual motion pulse experiments presented in Figures 2-4 were behavior-only experiments done as a direct follow-up to test our interpretation of the neural analyses of Figures 5-8.

First, we want to argue that this is not a glaring omission. The responses to a pulse of target motion at different times during fixation do not differ conceptually from the response to the onset of target motion at the end of the prescribed fixation period. Both initiate eye movement, both are driven by visual motion, and both represent neural correlates of the eye movement behavioral response. There is no reason to think that they would differ from the pursuit-related responses and the population activity during a probe should, by all accounts, align with the movement PCs (and only be partially reorganized from preparatory PCs). To collect these data would be a ton of work only to confirm a very solid prediction.

Second, we want to make an offer. A lab member who is not an author in this paper has collected neurophysiological data in FEFSEM during visual motion pulse experiments (n=51 neurons in one monkey). Even though there are some minor differences in the experimental design, we have analyzed those data and found a positive correlation (r=0.62) between preparatory modulation of firing rate and modulation of activity during the pulses (see Author response image 1). Thus, population modulation in response to the pulses is in the same direction as modulation during preparation. This is consistent with the relationship that we have shown between preparatory- and pursuit-related firing rates (Figure 1D) and weakly supports our expectation from doing this experiment correctly. We would be happy to include this as a supplementary figure and add the other lab member as an author on the paper if the reviewer deems that it materially adds to the story, even though we do not think it is a strong addition.

In summary, a study of how population activity relates to movement preparation and generation is timely. Tests of whether new hypotheses developed in skeletomotor systems generalize to the oculomotor system is very much needed, and this study takes a step in the direction. However, the analysis isn't rigorous enough to reach a reliable conclusion.

We appreciate the positive part of this comment. We have done the best we can to improve the rigor along the lines recommended by the reviewer, but we also respectfully disagree with one of the major comments concerning the use of dPCA. We agree that dPCA would be more “modern”, and have added dPCA in another portion of the manuscript. However, we think the analysis we have done in comparing preparatory and pursuit PCs is closer to unbiased and that this is a major virtue.

Reviewer #3:Darlington and Lisberger present a previously undescribed "mechanism" by which the smooth eye movement (SEM) region of FEF could prepare pursuit movements without causing them -by dialing up the visuomotor gain. Moreover, they provide evidence in the SEM system against the "null subspace" hypothesis of preparation of arm movements. I find many aspects of the manuscript interesting but I have some concerns about their analytical approach.Comments:1) Although I think that the potential differences in control of reaching movements and pursuits are interesting, I have concerns about how the authors tested the "null subspace" hypothesis. Overall, I think that the authors should replicate the methods proposed by Kaufman et al., rather than introduce a number of modifications whose implications are hard to grasp -at least for me.

We’d like to thank the reviewer for this feedback and agree that we did not justify/clarify our use of these modifications. The primary reason we modified our analytical approach is because there are key experimental differences between the way that we collected our data and the way in which the Kaufman et al. data was collected. We have added text to clarify. Furthermore, we have taken two major steps to remedy this concern. First, we have included control analyses to show that our conclusions are not due to the modifications. Second, we have included as Figure 6—figure supplement 3 a supplementary analysis that is more congruent with what was done in the Kaufman et al. paper.

In more detail:1a) All subspace analyses putatively capture activity that reflects some aspect of the underlying circuitry as well as the task (e.g, related to inputs to the neural population or outputs). However, the authors find the null subspace in a lower-dimensional version of the task that they probe (1-targets vs. 8-targets) that is also -I think- cued differently. My suggestion is that they do the analysis using the full 8-target data and considering the preparatory and pursuit epochs as Kaufman et al. did.

The only difference between the 1-direction and 8-direction tasks is the preponderance of target directions that occurs. There were no explicit cues in either task indicating the direction of the upcoming target motion. Therefore, accurate preparation is only possible during the 1-direction task where the monkey is able to anticipate the upcoming target motion. We decided to use the 8-direction task to perform the mapping of pursuit-related FEFSEM activity to eye movement because it provides a richer assessment of eye velocity. We have written more clearly about this issue, in a way that we hope resolves the reviewer’s concerns.

1b) Studies like Kaufman's assume that the relevant "neural information" is captured by activity patterns shared across the neural population; these patterns are probed by analyzing a state space in which axis is the activity of one recorded neuron. The authors adopted an approach that puzzled me: having multiple axes that reflect the same neuron after rotating its activity. I'd again suggest them to adopt the simpler approach of defining one axes per neuron, discarding neurons that were recorded multiple times during different conditions (at least I have trouble foreseeing all potential implications of their manipulations)1c) I am again a bit confused about the potential effect of the authors rotating the tuning curves of each neuron in the analysis. Although I agree that there may be FEF neurons with these properties, I would avoid it because it implies assuming that the absolute coordinates of the target doesn't matter, or at least show that it doesn't influence the main result.

We can understand how readers might be confused by some of the choices we used in our data analysis and we apologize for not being clearer in our explanations and for not including some obvious controls. The goal was to create as much of a population response as we could, without excluding data points or neurons and without biasing our data collection for the 1D experiment towards any part of the direction tuning curve of our recorded neurons (i.e. only running the 1D experiment in the preferred direction of our neurons). Therefore, we ran multiple single direction experiments (where preparation could effectively occur) for most neurons in different directions relative to their preferred direction. We then chose to rotate the direction tuning curves in a manner that made it seem like all of the 1D experiments were run in the same direction (0 degrees, rightward). This allowed us to construct a full pseudo-population preparatory response for which the full range of differences between the expected direction of the upcoming visual motion and the preferred direction of the cell is represented.

We have corrected our prior errors. Now, we include a control analysis where each neuron contributes a single data point and we show that our conclusions are unchanged (Figure 6—figure supplement 2). Furthermore, Figure 6—figure supplement 3 show that we obtain the same results when the analysis used only data from the 1D task and involved no rotation of neural data.

1d) Did the authors include all the recorded neurons in this analysis or only neurons that were significantly tuned to the task? Although studies in the motor cortices suggest that the population dynamics sampled from sufficiently large neural populations are the same irrespective of the sampled neurons (Trautman et al., 2018), I worry this may bias their results.

We included all recorded neurons in this analysis. We are now clear about this in the text.

1e) Because of these concerns, I believe that the authors should repeat Figure 8 after calculating the preparatory and pursuit PCs as in Kaufman et al., 2014.

We have now included as Figure 6—figure supplement 3 a supplementary analysis that is more congruent with that used in Kaufman et al., 2014 in that it uses data from only one task, computes a weight matrix between pursuit PCs and eye movement, and tests the projections of preparatory PCs onto the row and null spaces of the weight matrix. The results and conclusions are unchanged.

1f) Minor: I assume that the authors only compare the first pursuit PC and the first preparatory PC because they each capture a large fraction of the total neural variance. For a full assessment, I'd suggest an analysis like the one in Figure 4 of Elsayed et al., 2016, or the principal angle analysis used in Gallego et al., 2018.

We chose only the first PCs because they each captured ~70% of the variance, the same value used to choose the number of PCs used in Elsayed et al., 2016. We now state this clearly in the text.

2) The central idea of "gain control" is interesting, but I think the authors should pursue it further. For example, can single trial eye speed be predicted from neural activity? (I'd do this analysis for the data in Figure 2 and 3). Also, how do the authors think this gain leads to a transition from preparation to movement, by exceeding a threshold? I think some analyses and discussion in this regard are missing.

We have added some text to the Discussion to pursue our ideas about “gain control” further. A previous publication from our laboratory showed that single trial eye speed can be predicted from neural activity during the initiation of pursuit (Schoppik et al., 2008). Because Figures 2 and 3 show the results of behavioral experiments without neural recordings, we cannot address this cogent question with the data at hand. We would, however, point out that single-trial neuron-behavior correlations are more of a statement about neuron-neuron correlations than about any particular functional role and, so, we view that question as tangential to the main point of the present paper. In terms of what triggers movement, we have just published a paper that shows the absence of a threshold for triggering movement based on the signals in the FEFSEM (Lee et al., 2019). Instead, we think that the gain signal from FEFSEM poises the system for action, but movement is triggered only by visual motion signals that are subject to the gain modulation.

3) In the Results section the authors make the proposal that there may be "non-specific preparatory enhancement of visuomotor gain". I think they could address this directly by using demixed principal component analysis (dPCA) (Kobak et al., 2016). dPCA effectively performs semi-supervised dimensionality reduction and can be used to isolate population activity patterns that only depend on time (these would be the component the authors refer to) and components that depend on the direction of the movement (see, e.g., Kaufman et al., 2016, Gallego et al., 2018 for examples in motor cortex). Examining both the dynamics of these components (e.g., when do putative time-related and target-related components start ramping up?) and the weights (e.g., are there subpopulations of neurons?) could be interesting. Note that there's freely available code by Kobak et al.

We’d like to thank the reviewer for this excellent suggestion. We used dPCA on a subset of our FEFSEM population in which we were able to collect single-direction data both toward and away from the preferred direction of each neuron. The results are shown in Figure 4 and reveal a very nice correlate with the behavioral findings of Figure 4A, making an even stronger link between FEFSEM preparatory activity and the preparatory modulation of visual-motor gain.

4) Figure 2 shows strong effects, but it would be nice to see the data distributions, or at least box plot versions of (b) and (c). As to (d), would the authors see the same trend if they plotted single trial values or a 2D histogram? This figure focuses in neurons that had "positive preparatory firing rate modulation". What was their percentage among all recorded neurons? And among all modulated neurons? And what happens with these neurons? Same for the increase and decrease cells in Figure 3.

The source data for Figure 2 (a and b) panels are included with the submission. We have added information to the paper clarifying the percentage of neurons categorized as having positive versus negative preparatory firing rate modulation. The behavioral data for the pulses was collected separately (though under identical conditions) from the neurophysiology data. Therefore, it is not possible to do this analysis with single trials. However, our prior papers (Darlington et al., 2018; Lee et al., 2019) show that there is a positive relationship between fixation time and preparatory firing rate, and unpublished data show a correlation between preparatory firing rate and ensuing pursuit speed on a trial-by-trial basis. We would be willing to include the latter data in Figure 2, but it seems somewhat tangential to the main points of this paper.

[Editors' note: further revisions were suggested prior to acceptance, as described below.]

We have one remaining major concern that should be addressed:1) It would be interesting to a) recompute weights relating FEFsem activity to eye velocity separately for subpopulations 1 and 2, and (b) predict eye velocities for each. Is there a difference in the predictive power for the two subpopulations? And how well do the results align with the proposed role of each subset (including for different directions)? A new figure can probably be committed to the results.

We performed the linear model analysis from Figures 5 and 6 separately on the two subpopulations. This analysis has been added to the manuscript as Figure 9. It reveals that both subpopulations perform roughly the same in terms of predicting the eye velocity during pursuit in the single-direction block. Furthermore, it shows that preparatory activity in subpopulation 1 evolves in a way that predicts movement in the direction of the upcoming visual motion and eye movement. However, preparatory activity from subpopulation 2 predicts no movement on average.